# MSEarth: A Multimodal Scientific Dataset and Benchmark for Phenomena Uncovering in Earth Science

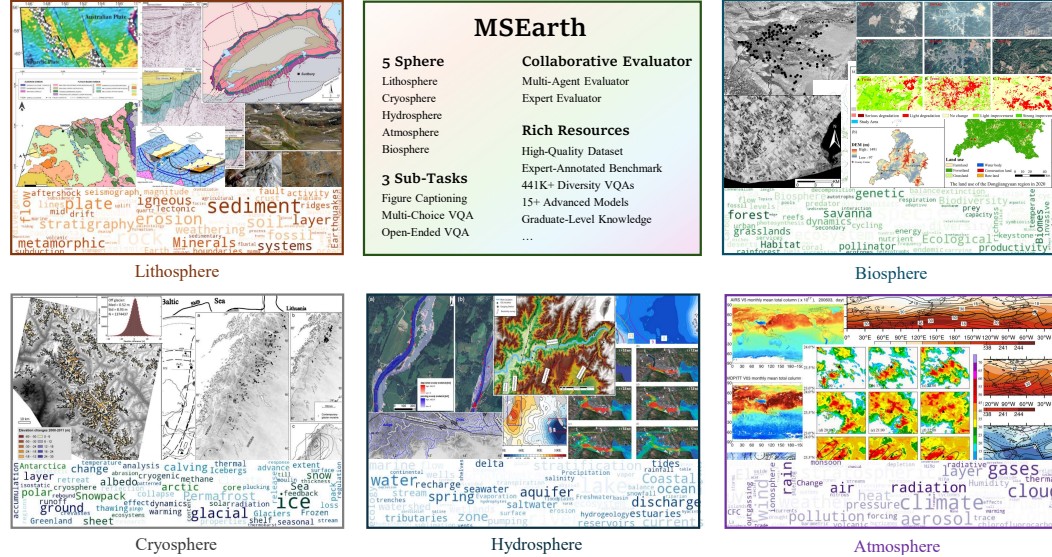

Figure 1: Illustrative examples of the diverse types of scientific figures in MSEarth, sourced from open-access articles available from website.

## Abstract

The rapid advancement of multimodal large language models (MLLMs) has unlocked new opportunities to tackle complex scientific challenges. Despite this progress, their application in addressing earth science problems, especially at the graduate level, remains underexplored. A significant barrier is the absence of benchmarks that capture the depth and contextual complexity of geoscientific reasoning. Current datasets and benchmarks often rely on synthetic datasets or simplistic figure-caption pairs, which do not adequately reflect the intricate reasoning and domain-specific insights required for real-world scientific applications. To address these gaps, we introduce MSEarth, a multimodal scientific dataset and benchmark curated from high-quality, open-access scientific publications. MSEarth encompasses the five major spheres of Earth science: atmosphere, cryosphere, hydrosphere, lithosphere, and biosphere, featuring over 289K figures with refined captions. These captions are crafted from the original figure captions and enriched with discussions and reasoning from the papers, ensuring the benchmark captures the nuanced reasoning and knowledge-intensive content essential for advanced scientific tasks. MSEarth supports a variety of tasks, including scientific figure captioning, multiple choice questions, and open-ended reasoning challenges. By bridging the gap in graduate-level benchmarks, MSEarth provides a scalable and high-fidelity resource to enhance the development and evaluation of MLLMs in scientific reasoning. The benchmark is publicly available to foster further research and innovation in this field. Resources related to this benchmark can be found at this anonymous link: https://anonymous.4open.science/r/MSEarth-2B3F.

# 1 INTRODUCTION

The advent of multimodal large language models (MLLMs) (Liu et al., 2023a; Liang et al., 2024b) has revolutionized artificial intelligence, driving groundbreaking advancements across diverse scientific disciplines. Prominent examples include ChemVLM (Li et al., 2025) in chemistry, GeoChat (Kuckreja et al., 2024) in geography, and WeatherQA (Ma et al., 2024) in atmospheric science. These models excel in domain-specific visual question answering by integrating specialized knowledge. For example, ChemVLM facilitates the analysis of molecular structures, chemical reactions, and chemistry-related examination questions, while WeatherQA enables reasoning about severe weather events in real-world scenarios.

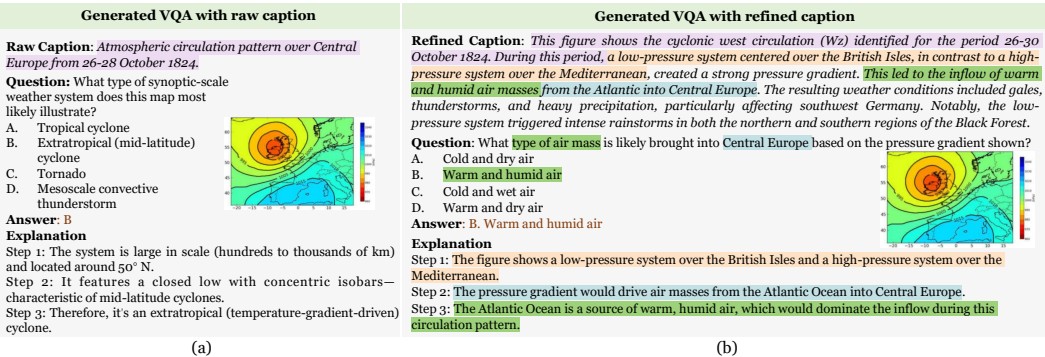

Figure 2: Illustration of VQA generation methodologies: (a) VQA relying exclusively on figure captions, and (b) VQA utilizing refined captions that integrate figure captions with content from academic papers. Highlighted areas denote questions and answers supported by evidence.

Developing multimodal large language models (MLLMs) that understand advanced geoscientific knowledge necessitates rigorous datasets and benchmarks (D&Bs) to enhance their ability to solve complex, discipline-specific problems. Existing datasets and benchmarks often rely on synthetic datasets or materials from high school and undergraduate textbooks (Lu et al., 2022; Yue et al., 2024), which lack the depth required for professional, graduate-level tasks. Recent efforts (Li et al., 2024c; Roberts et al., 2024; Li et al., 2024b) have turned to academic papers for constructing multimodal scientific benchmarks, leveraging the complexity of graduate-level content. Yet, these approaches typically extract only figures and captions, neglecting the critical scientific reasoning and insights in the paper context. Consequently, tasks based on these benchmarks tend to be oversimplified, focusing on basic *figure-caption matching*, which offers limited insight into a model's reasoning capabilities. Another obstacle to advancing current datasets and benchmarks is the difficulty of designing questions that rigorously evaluate MLLMs' data-analysis abilities to uncover Earth-science phenomena from observational imagery. In scientific papers, images depicting observed phenomena are often accompanied by information that reveals scientific hypotheses, supporting evidence, analyses, and conclusions—critical insights that are typically embedded within the main text rather than the figure captions alone. As shown in Figure 2 (a), question generation based solely on captions oversimplifies tasks, as the quality and difficulty of questions are constrained by the limitations of the generation models, and they lack the contextual support from the papers, making verification challenging. Existing benchmarks focus solely on figure-caption matching while neglecting the high knowledge density in scientific reasoning processes. This raises a new challenge: *How to effectively align high-value Earth science images with long-context information for phenomena uncovering?*

To address this challenge, we propose a novel approach to D&B construction that overcomes the limitations of existing methods through two key innovations. First, we introduce the concept of the *refined caption*. In scientific papers, observational images typically serve as visual representations of phenomena, while the deeper scientific insights—such as hypotheses, supporting evidence, analytical reasoning, and conclusions—are often embedded within the main body of the text. Raw captions provide only brief descriptions and lack the necessary context for complex reasoning. The refined caption bridges this gap by combining the raw caption with relevant, domain-specific information extracted from the paper, resulting in a more comprehensive and scientifically meaningful description

of the image. As illustrated in Figure 2 (b), questions generated using refined captions not only exhibit higher quality but are also supported and validated by the content of the academic papers, ensuring the professionalism and accuracy of the generated questions. Second, we implement a rigorous quality control process to validate the generated benchmark. This process combines a multi-agent automated evaluation and expert evaluation, ensuring that the generated questions are relevant, coherent, and aligned with the complexity of professional-level geoscientific reasoning. By addressing the reliability and accuracy issues faced by current LLMs in scientific question generation, our approach ensures that the benchmark is both robust and valuable for evaluating MLLMs in advanced scientific domains.

Building on our adaptive annotation methodology, we present MSEarth, a comprehensive multimodal D&B for graduate-level Earth science. This D&B is derived from 64,560 open-access publications spanning five spheres, eight subjects, and 66 sub-subjects, from which we extract 289,891 figures. We enhance these figures with "refined captions" that average 136.29 tokens in length, significantly expanding from the original average of 37.56 tokens. The test set unifies scientific figure captioning with multiple-choice and open-ended reasoning tasks, providing a holistic, interdisciplinary evaluation framework. This rigorously assesses MLLMs within professional-level geoscientific contexts, filling a critical gap in graduate-level multimodal benchmarks. Furthermore, MSEarth introduces a scalable, high-fidelity pipeline for constructing domain-specific scientific D&B, establishing the test set as a robust tool for advancing MLLMs in real-world Earth science applications. Leveraging this framework, we produce a resource-rich training dataset covering captioning, open-ended QA, and multiple-choice tasks for post-training (e.g., instruction tuning and GRPO-based reinforcement learning). This training data delivers substantial improvements across a wide range of tasks for open-source models. In summary, our contributions are as follows:

**Development of a Scalable Adaptive Framework:** We introduced an innovative semi-automated tool, enabling the automatic generation and machine-assisted filtering of VQA tasks. This provides a robust, scalable solution for creating high-fidelity, domain-specific D&B that can be extended to other scientific fields.

**High-Quality D&B Resources for Earth Science MLLMs:** We present an expert-annotated benchmark for graduate-level Earth science, along with a diverse training corpus—including captioning, open-ended QA, and multiple-choice tasks—to support advanced post-training of MLLMs.

**Comprehensive Evaluation and Validation of State-of-the-Art MLLMs:** Through extensive evaluations of MLLMs on MSEarth, we not only provide crucial insights into current limitations and future research directions but also demonstrate the effectiveness of our training data by developing a state-of-the-art baseline model.

## 2 RELATED WORK

**Multimodal Scientific Datasets and Benchmarks.** Numerous multimodal D&Bs have been developed to evaluate scientific understanding across various domains. These benchmarks often integrate text, images, and other modalities to assess models' reasoning and cross-modal capabilities. However, their creation typically requires significant manual effort in data collection and validation. ScienceQA (Lu et al., 2022) is an early multimodal benchmark that features multiple-choice questions (MCQs) collected from online resources and manually filtered for quality. It covers general science topics such as physics, chemistry, and biology, with a focus on elementary and high school-level reasoning. SceMQA (Liang et al., 2024a) and Mmmu (Liang et al., 2024a) extended this by incorporating both MCQs and open-ended questions (OE) from textbooks and online resources, targeting pre-college and college-level difficulty. OlympiadBench (He et al., 2024a) introduced competition-level problems in mathematics and physics, offering open-ended tasks sourced from Olympiad exams. These problems are highly challenging but limited to specific domains. More recently, EMMA (Hao et al., 2025) combined manually designed questions with existing benchmarks, covering a broader range of topics with mixed difficulty levels. In contrast, our objective is to enhance models' ability to comprehend multimodal, complex scientific problems—drawn from high-quality research papers in earth science—that demand graduate-level, domain-specific expertise.

**Paper-Based Multimodal Scientific Datasets and Benchmarks.** D&Bs based on academic papers aim to leverage the rich, domain-specific content found in scientific literature. FigureSeer (Siegel

Figure 3: Data curation process for MSEarth. The two parts on the left represent data preprocessing, while the two parts on the right encompass the automated generation of VQA and expert-AI collaborative filtering.

et al., 2016) first extracts figures from academic papers, focusing on chart figures to evaluate the understanding of chart figures. SciFiBench (Roberts et al., 2024) extended this by introducing figure-to-caption and caption-to-figure matching tasks, while MMSci (Li et al., 2024c) further advanced this approach using figures from Nature papers. However, these benchmarks lack original questions, limiting their ability to assess advanced reasoning and contextual understanding. ArxivQA/Cap (Li et al., 2024b) expanded the scope by generating new questions for figures from 32 subjects on arXiv. However, these questions were generated solely using the inherent capabilities of GPT-4V (Achiam et al., 2023) and did not have contextual support from the relevant text in the papers, raising concerns about their scientific validity. In contrast, our proposed benchmark, MSEarth, addresses these limitations by introducing original, evidence-supported questions grounded in refined captions. This approach enables a rigorous evaluation of MLLMs in professional-level geoscientific applications.

# 3 MSEARTH - A MULTIMODAL SCIENTIFIC D&B FOR EARTH SCIENCE

This section provides a detailed overview of the construction process for MSEarth. As illustrated in Figure 3, we outline the framework used to develop MSEarth from open-access scientific publications. The section is organized into three main parts: first, we detail the data collection and preprocessing steps. Next, we elaborate on the construction procedures for the two D&B within MSEarth, namely MSEarthCap and MSEarthQA. Finally, we describe the process of ensuring the reliability of the test data, which involves expert annotation and manual screening of the sampled test data.

## 3.1 DATA PREPARATION

The first part of the D&B construction focuses on data collection and preprocessing. The data collection begins with more than 400K Earth science papers obtained in PDF format. These are uniformly converted into structured JSON text using the MinerU (Wang et al., 2024) parser. To classify the papers, semantic similarity is calculated between the abstracts and keywords from the five Earth spheres: hydrosphere, biosphere, lithosphere, atmosphere, and cryosphere. Based on this, the papers are assigned to respective disciplinary categories. Details are provided in Appendix B.3. We then selected papers based on the criterion of containing high-quality, Earth science-related images, resulting in a subset of around 83k papers. Specifically, Qwen-2.5-VL-72B (Bai et al., 2025b) is utilized to filter and select images, with the filtering prompts detailed in the Appendix B.4.

## 3.2 MSEARTHCAP

**Figure-Caption Extraction**: Figures and their corresponding captions are extracted from the JSON files processed by MinerU. As shown in Appendix B.2, MinerU has already extracted the figures along with their original captions, which can be directly utilized for subsequent processing. To ensure accurate alignment between figures and their references within the text, we employ a regex-based method to identify the labels of each figure. This approach enables precise matching between the figures and the relevant sections of the articles.

Figure 4: Overall approach of our multi-agent, voting-based approach to automate the validation of generated questions.

**Relevant Context Extraction**: To further enrich the captions with contextual information, we use the figure labels obtained in the previous step to perform approximate matching against the main body of the paper. Since MinerU processes papers with segmented paragraphs, we apply regular expression matching to each paragraph to extract contextual text that references the target figure. This ensures the inclusion of descriptions and reasoning associated with each figure within the paper. To guarantee that the extracted context provides sufficient detail about the target figure, only paragraphs exceeding two sentences were included in the final dataset. From this filtered subset, we selected around 64K papers that met the criteria for subsequent processing. For more details, refer to Appendix B.5.

**Refined Caption Generation**: To create professional-level figure descriptions, we employ GPT-4o for refined caption generation. The model takes as input the extracted figure, its original caption, and the contextual text from the relevant sections of the paper. Figure refinement is performed only for data that includes valid relevant contexts. The specific prompts used for this process are detailed in the Appendix B.6. After statistical analysis, we observe that the average word length of the raw captions is 37.56, while the average length of the refined captions increases to 136.29, reflecting the incorporation of richer, domain-specific content.

## 3.3 MSEARTHQA

To generate high-quality multiple-choice questions (MCQs) and open-ended questions, we use a question generation pipeline that takes the figure, its original caption, and the refined caption as input. The generation prompts, detailed in the Appendix B.6, are crafted to encourage the model to highlight differences between the original and refined captions, ensuring that the generated questions are grounded in evidence from the paper. The questions are constructed using GPT-4o to maintain high detail and relevance. However, due to inherent challenges such as self-inconsistency and uncertainty in the generation process, the generated questions undergo an automated and expert validation process to ensure quality.

### 3.3.1 AUTOMATED VALIDATION

Inspired by the LLM Voting (Yang et al., 2024b; Lee et al., 2025; Kaesberg et al., 2025) method, we developed a multi-agent, voting-based approach to automate the validation of generated questions. Specifically, we employ a *Majority Voting* strategy, where multiple agents independently generate responses, and the final decision is based on the majority consensus of these agents. In our setup, we utilize the following MLLMs for decision-making: Qwen2.5-VL-72B, Qwen2.5-VL-7B, InternVL2.5-7B, InternVL2.5-78B, and GPT-4o. A key aspect of our evaluation process is the use of the *refined caption*, which incorporates scientists' reasoning and insights about the figure extracted from the paper. This refined caption provides additional context and domain-specific information that goes beyond the original caption. By comparing model performance with and without the refined caption, we can assess the quality of the questions and determine whether they test the model's ability to grasp scientific reasoning and insights. The detailed decision-making process is outlined below:

**Phase A:** The question and original caption are provided to a suite of models $\{M_1, M_2, ..., M_n\}$. The correctness of the model responses is used to evaluate the types and quality of the questions. A threshold of 60% is defined for supermajority voting. Specifically, if more than 40% of the models produce incorrect responses, the question is flagged for further analysis. Additionally, we discard questions that all models answer correctly, as these questions do not contribute to the effective testing or training of the models. Questions identified through this process are categorized as either potentially difficult or of poor quality, with the distinction made in subsequent phases.

**Phase B:** In this phase, the question and refined caption are provided to the same suite of models. If more than 60% of the models answer the question correctly with the refined caption, it indicates that the question requires relatively specialized scientific knowledge to answer. Such questions are categorized as *specialized QA*, as their answers rely on the model's ability to understand and apply specific domain knowledge rather than simply perceiving the image or relying on commonsense reasoning. Questions that fail this phase proceed to the next stage for further evaluation.

**Phase C:** In this phase, only models with 70B+ parameters are used for voting, including GPT-4o (the same model used for question generation), InternVL2.5-78B, and Qwen2.5-VL-72B. The question and its refined caption are provided to these large-scale models. If more than 60% of the large models answer the question correctly, it suggests that the difficulty of the question likely lies in the model's ability to perceive and interpret the image content. Such questions are categorized as *hard QA*. Subsequent human validation will involve sampling and additional annotation across QA filtered in all phases to ensure the overall quality and accuracy of the benchmark.

This pipeline identifies high-quality questions by filtering out overly simplistic or poorly constructed ones. In **Phase A**, approximately 70% of the questions were categorized as *easy*, as most models could answer them correctly without refined captions. After **Phase B**, around 20% were classified as *specialized QA*, where refined captions enabled correct answers, indicating the need for domain-specific knowledge. In **Phase C**, 5% were labeled as *hard QA*, requiring high-performing models to interpret image content accurately, suggesting that these questions test the model's ability to perceive and interpret image content. The remaining 5% were deemed flawed and discarded. Detailed processes and examples are provided in Appendix C. For the training dataset, we sampled more than 150K VQA pairs, including both multiple-choice and open-ended questions. Of these, 20% were drawn from Phase A and 80% from Phase B, ensuring a balanced distribution of question difficulty.

Table 1: Main statistics in MSEarth-Bench.

| Statistic | Number |
|---|---|
| Total subjects | 66 |
| Total articles | 64,560 |
| Total figures | 289,891 |
| Total questions | 448,980 |
| Training Set | 441,785 |
| Captioning as QA | 289,891 |
| MCQ | 102,753 |
| Open-Ended QA | 49,141 |
| Test Set | 7,195 |
| Captioning as QA | 3,000 |
| MCQ | 2,784 |
|   * Reasoning Question | 2117 (76.0%) |
|   * Perception Question | 667 (24.0%) |
| Open-Ended QA | 1,411 |
| Average caption length | 37.56 |
| Average refined caption length | 136.29 |

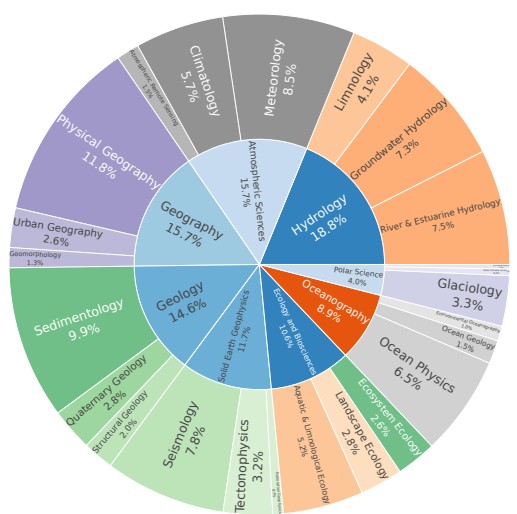

Figure 5: Subjects distribution in MSEarth.

### 3.3.2 EXPERT VALIDATION

Ensuring that synthetic data closely mirrors real-world distributions is critical for evaluation tasks. To achieve this, domain experts are engaged to review and annotate the curated QA pairs for accuracy and relevance. During this process, low-quality or invalid questions are identified and filtered out to ensure the overall quality of the dataset. The annotation process is conducted from two perspectives: image types and question types. For image types, we categorize the data into three groups: single-image question answering, single-image-focused question answering within multi-image figures, and multi-image relational question answering. For question types, we define two categories: scientific reasoning and perception questions. Scientific reasoning questions are constructed based on inferences or scientific discoveries presented in research papers, making them more specialized and challenging. In contrast, image perception questions focus on interpreting images and require less background knowledge of scientific concepts. This expert-AI collaborative process, combined with rigorous quality control, results in a high-quality dataset comprising 1,500 open-ended questions and 3,000

MCQs, forming the MSEarthQA benchmark. The annotated results are summarized in Table 1, with further details and analysis provided in Appendix D and Table 10.

# 4 EXPERIMENTS

## 4.1 EVALUATED MODELS

We evaluate different families of MLLMs on our benchmark. We evaluate the following closed-source models: GPT-4 series (Hurst et al., 2024), Gemini-2.5 series (Team et al., 2023) and Claude-3 series (Anthropic, 2024). We also evaluate the following open-source models: LLaVA-OneVision (Li et al., 2024a), Qwen-2.5-VL (Yang et al., 2024a), InternVL2.5/3/S1 (Chen et al., 2024; Zhu et al., 2025; Bai et al., 2025a) and Llama-3.2-Vision-Instruct (Grattafiori et al., 2024). We use chat/instruction-tuned variants of each model and compare the performance of multiple model sizes where available. Details can be found in Appendix F. To validate the effectiveness of our training data, we conducted post-training on the Intern-s1-mini and Qwen-2.5-VL-7b models. Specifically, we employed instruct tuning method for fine-tuning on captioning and open-ended QA tasks. For the MCQ task, we applied GRPO (Shao et al., 2024) reinforcement learning method.

## 4.2 EVALUATION METRICS

Both captioning and open-ended QA tasks require generating freeform textual outputs grounded in complex scientific data and reasoning. To evaluate these tasks, we use lexical overlap-based metrics such as BLEU (Papineni et al., 2002), ROUGE (Lin, 2004), and METEOR (Banerjee & Lavie, 2005) for surface-level similarity, while BERTScore (Zhang et al., 2019) assesses deeper semantic alignment. Given the importance of factual correctness, we employ a factual entailment classifier to measure the consistency of generated outputs with reference answers or captions. Additionally, following G-Eval (Liu et al., 2023b), we utilize the Qwen2.5-VL-72B model with a specialized prompt to compute a factual scientific score. For the captioning task, we define a Cap-Eval score ranging from 1 to 5, where higher scores indicate better caption quality. For the open-ended QA task, we introduce OE-Eval, which evaluates the reasonableness of generated answers using a binary 0/1 scoring system.

Evaluating MCQs is relatively straightforward, as these tasks require selecting the correct answer from a predefined set of options. In our experiments, models were guided by carefully structured prompts to ensure adherence to a specific output format. Regular expression rules were employed to extract the selected choice, ensuring strict alignment with the predefined answer format. We observed that some models occasionally failed to consistently follow the formatting instructions. To address this issue and preserve the integrity of the evaluation process, we implemented a similarity-based method to identify the closest matching option when the selected choice could not be extracted using regular expressions. Detailed evaluation metrics are provided in Appendix G.

## 4.3 MAIN RESULTS

**Current models struggle with scientific question-answering tasks, particularly on questions requiring specialized knowledge and reasoning across multiple images.** The performance of MCQs is summarized in Table 2. The results reveal that most models do not perform exceptionally well on scientific question-answering tasks, with proprietary models generally achieving better results. Further analysis of the models' failure rates on reasoning and perception-based questions is provided in the Appendix (Figure 8). This analysis shows that models are more prone to errors on questions requiring specialized knowledge, underscoring significant room for improvement in scientific reasoning question-answering. In contrast, for relatively simpler perception-based questions, which require less domain-specific knowledge, the models tend to perform better. Similarly, when analyzing performance across different image types, we observe that most models achieve their best results on tasks involving single-image inputs. However, for tasks requiring multi-image inputs, particularly those that demand reasoning across multiple images to derive an answer, the models perform the worst. Additional experimental results can be found in Appendix H.

**Proprietary models consistently outperform open-source models in both Scientific Figure Captioning and Open-Ended VQA tasks, with LLM-based metrics providing a more reasonable**

Table 2: Accuracies (%) of different models on multiple-choice questions. The best results are highlighted in bold, with the second-best underlined.

| Model | Image-Type | | | Task Type | | Overall |
| | SINGLE | MULTI | CROSS | REASONING | PERCEPT | ACC |
|---|---|---|---|---|---|---|
| *Open-source Models* | | | | | | |
| LLaVA-onvision-72B | 53.55 | 49.48 | 47.95 | 46.58 | 65.52 | 51.11 |
| Qwen2.5-VL-7B | 47.65 | 44.07 | 37.53 | 40.53 | 58.47 | 44.83 |
| Qwen2.5-VL-32B | 52.59 | 46.99 | 43.84 | 42.47 | 70.16 | 49.10 |
| Qwen2.5-VL-72B | 52.11 | 50.43 | 46.30 | 44.40 | 70.46 | 50.65 |
| InternVL2-8B | 44.86 | 43.99 | 38.36 | 38.97 | 58.47 | 43.64 |
| InternVL2.5-78B | 53.23 | 49.74 | 44.38 | 43.17 | 74.21 | 50.61 |
| InternVL3-78B | 57.53 | 51.37 | 45.48 | 47.00 | 73.61 | 53.38 |
| Llama3.2-90B-Vision | 45.98 | 40.46 | 38.90 | 38.26 | 56.97 | 42.74 |
| DeepSeek-VL2 | 52.43 | 49.23 | 44.66 | 46.06 | 62.82 | 50.07 |
| Intern-S1-mini | 60.00 | 57.22 | 49.04 | 51.68 | 75.56 | 58.12 |
| Intern-S1 | 67.01 | 65.62 | 64.11 | 61.22 | 79.61 | 65.63 |
| MiMo-VL-7B-RL | 51.31 | 47.51 | 32.33 | 42.70 | 61.62 | 47.23 |
| GLM-4.1V-Thinking | 54.18 | 49.83 | 38.36 | 46.29 | 62.97 | 50.29 |
| Qwen-7B-MSEarth | 57.61 | 52.75 | 45.20 | 50.68 | 64.32 | 53.95 |
| Intern-S1-mini-MSEarth | **65.49** | **62.97** | **58.63** | **58.81** | 78.56 | **63.54** |
| *Proprietary Models* | | | | | | |
| Gemini-2.5-Flash | 58.33 | 54.55 | 53.42 | 49.98 | 75.56 | 56.11 |
| Gemini-2.5-Flash-Thinking | 60.64 | 54.64 | 53.70 | 51.35 | 75.86 | 57.22 |
| Gemini-2.5-Pro-Thinking | 64.78 | 59.36 | 55.34 | 56.31 | 77.06 | 61.28 |
| Claude-3.5-Haiku | 49.48 | 47.16 | 42.47 | 42.18 | 64.77 | 47.59 |
| Claude-3.7-Sonnet | 59.52 | 56.53 | 57.53 | 51.68 | 78.11 | 58.01 |
| GPT-4o-mini | 52.51 | 48.63 | 43.01 | 43.65 | 68.67 | 49.64 |
| GPT-4o | 63.03 | 55.76 | 47.67 | 50.45 | **81.86** | 57.97 |

Table 3: Performance on scientific figure captioning. The best results are highlighted in bold, with the second-best underlined.

| Model | | | Overall | | | Similarity | MLLM |
| | ROUGE1 | ROUGE2 | ROUGEL | METEOR | BLEU | BERTSCORE | CAP-EVAL |
|---|---|---|---|---|---|---|---|
| *Open-source Models* | | | | | | | |
| LLaVA-onevision-72B | 30.81 | 5.99 | **17.82** | 18.35 | 2.15 | 83.47 | 2.07 |
| Qwen2.5-VL-7B | 27.01 | 5.21 | 15.72 | 16.57 | 1.55 | 83.82 | 2.22 |
| Qwen2.5-VL-32B | 29.90 | 5.88 | 15.53 | 25.81 | 2.14 | 82.86 | 2.66 |
| Qwen2.5-VL-72B | 30.68 | 5.85 | 16.53 | 21.35 | 2.36 | 83.87 | 2.56 |
| InternVL2.5-8B | 29.07 | 5.16 | 16.71 | 19.28 | 1.58 | 83.25 | 1.91 |
| InternVL2.5-78B | 30.66 | 5.95 | 17.24 | 20.66 | 2.27 | 83.56 | 2.30 |
| InternVL3-78B | 30.73 | 5.87 | 16.95 | 20.95 | 2.32 | 83.72 | 2.43 |
| Intern-S1-mini | 31.05 | 7.13 | 17.42 | 24.18 | 2.26 | 83.65 | 2.65 |
| Llama3.2-90B-Vision | 19.93 | 4.32 | 12.98 | 21.21 | 1.49 | 78.89 | 1.82 |
| DeepSeek-VL2 | 29.06 | 5.42 | 16.71 | 18.22 | 1.69 | 83.64 | 2.22 |
| Intern-S1 | 33.46 | 7.37 | 17.93 | 27.97 | 3.29 | 83.95 | 3.41 |
| MiMo-VL-7B-RL | 29.84 | 5.72 | 15.24 | 25.18 | 2.11 | 83.15 | 2.61 |
| Qwen-7B-MSEarth | 32.36 | 7.33 | 17.74 | 26.59 | 2.86 | 83.71 | 2.73 |
| Intern-S1-mini-MSEarth | 33.41 | 7.64 | **17.94** | **27.61** | 3.25 | 83.86 | 3.02 |
| *Proprietary Models* | | | | | | | |
| Gemini2.5-Flash | 32.89 | 7.30 | 17.79 | 23.01 | 3.10 | **83.96** | 2.98 |
| Gemini2.5-Flash-Thinking | 32.67 | 7.00 | 17.42 | 23.47 | 2.97 | 83.85 | 3.04 |
| Gemini2.5-Pro-Thinking | **33.47** | **7.69** | 17.15 | 27.45 | **3.33** | 83.62 | **3.35** |
| Claude-3.5-haiku | 26.63 | 4.47 | 14.97 | 17.58 | 1.38 | 83.46 | 2.36 |
| Claude-3.7-Sonnet | 29.89 | 5.52 | 15.94 | 21.15 | 2.15 | 83.62 | 2.71 |
| GPT-4o-mini | 28.39 | 5.00 | 15.96 | 18.45 | 1.59 | 83.90 | 2.40 |
| GPT-4o | 29.89 | 5.57 | 16.33 | 20.19 | 2.15 | 83.93 | 2.72 |

**evaluation.** The captioning results are presented in Table 3, where overlap-based, similarity-based, and LLM-based metrics exhibit similar trends, with no significant differences observed among the overlap-based and similarity-based metrics. The Gemini-2.5-Pro model achieves the best performance across most metrics. The LLM-based metric, designed to evaluate the professionalism and accuracy

of generated captions, demonstrates greater variance compared to similarity-based metrics, making it more suitable for assessing the Scientific Figure Captioning task. Open-source models still show a noticeable gap compared to proprietary models, consistent with the findings from the MCQ results, suggesting a close interconnection between a model's understanding and reasoning capabilities. Similarly, the results for open-ended question answering, presented in Table 4, show that overlap-based and similarity-based metrics tend to yield higher scores due to the shorter nature of both ground truth answers and model-generated responses. However, for open-ended questions, the focus should be on the rationality and correctness of the answers, making the LLM-based metric a more reasonable evaluation method. This metric also reveals trends consistent with the previous tasks, further highlighting the performance gap between open-source and proprietary models.

Table 4: Performance on scientific open-ended question answering. The best results are highlighted in bold, with the second-best underlined.

| Model | Overlap | | | | | Similarity | LLM |
|---|---|---|---|---|---|---|---|
| | ROUGE1 | ROUGE2 | ROUGEL | METEOR | BLEU | BERTSCORE | OE-EVAL(%) |
| *Open-source Models* | | | | | | | |
| LLaVA-onevision-72B | 38.07 | 20.62 | 37.91 | 27.88 | 1.94 | 89.72 | 41.56 |
| Qwen2.5-VL-7B-Chat | 35.52 | 20.71 | 35.24 | 29.00 | 2.40 | 88.62 | 40.68 |
| Qwen2.5-VL-32B-Chat | 36.82 | 21.33 | 36.51 | 29.33 | 1.83 | 89.20 | 41.96 |
| Qwen2.5-VL-72B-Chat | 38.57 | 23.49 | 38.34 | 30.65 | 2.20 | 89.22 | 44.82 |
| InternVL2.5-8B | 36.12 | 21.05 | 35.95 | 28.25 | 2.04 | 89.14 | 39.05 |
| InternVL2.5-78B | 40.59 | 24.14 | 40.34 | 31.15 | 2.23 | 90.05 | 45.64 |
| InternVL3-78B | 40.59 | 23.87 | 40.42 | 31.77 | 2.37 | 89.98 | 47.00 |
| Llama3.2-90B-Vision | 37.64 | 22.26 | 37.53 | 29.08 | 1.99 | 89.16 | 42.72 |
| DeepSeek-VL2 | 36.49 | 20.44 | 36.37 | 27.24 | 1.83 | 89.38 | 40.68 |
| Intern-S1-mini | 36.43 | 22.04 | 37.46 | 29.72 | 2.34 | 89.25 | 43.69 |
| Intern-S1 | 43.24 | 22.04 | 37.46 | 29.72 | 2.34 | 89.25 | 43.69 |
| GLM-4.1V-Thinking | 3 | 22.04 | 37.46 | 29.72 | 2.34 | 89.25 | 43.69 |
| Qwen-7B-MSEarth | 39.55 | 22.32 | 38.47 | 30.10 | 2.37 | 89.82 | 48.19 |
| Intern-S1-mini-MSEarth | **41.82** | **25.11** | **41.36** | 31.87 | 2.44 | **90.42** | 49.74 |
| *Proprietary Models* | | | | | | | |
| Gemini-2.5-Flash | 40.26 | 22.98 | 40.02 | 32.34 | 2.06 | 89.39 | **52.00** |
| Gemini-2.5-Flash-Thinking | 39.59 | 22.30 | 39.47 | 30.77 | 1.87 | 89.58 | 46.49 |
| Gemini-2.5-Pro-Thinking | 38.67 | 22.30 | 38.38 | 31.73 | **2.50** | 88.93 | 47.70 |
| Claude-3.7-Sonnet | 40.61 | 23.58 | 40.21 | 30.75 | 1.73 | 89.37 | 48.33 |
| GPT-4o-mini | 36.49 | 21.37 | 36.27 | 28.89 | 1.63 | 89.06 | 41.81 |
| GPT-4o | 41.30 | 24.74 | 41.03 | **32.70** | 2.04 | 89.78 | 48.55 |

### 4.3.1 ANALYSIS AND DISCUSSION

As shown in Table 2, most MLLMs exhibit a pronounced gap between perception-based and reasoning-based performance. On simple visual questions—where answers depend on direct feature extraction—these models routinely exceed 75% accuracy. However, on scientific reasoning tasks that demand domain-specific knowledge, their scores drop sharply (e.g. Gemini-2.5-Pro-Thinking: 77.06% perception vs. 56.31% reasoning). This divergence suggests that while robust perception is a necessary foundation, it is not sufficient for Earth-science inference: once a model surpasses the $\approx 75\%$ perception threshold, further gains hinge on integrating specialized knowledge and enabling multi-step reasoning. By contrast, models pretrained or fine-tuned on scientific datasets, such as Intern-S1, demonstrate substantially higher accuracy in both perception and reasoning, thereby confirming that general MLLMs lack the requisite Earth-science expertise. Furthermore, we find that further training on the MSEarth training set boosts performance across the board, with the largest relative improvement appearing in reasoning tasks. Taken together, these results underscore the critical role of domain-focused data and architectures in closing the reasoning gap.

Key factors driving low reasoning-task performance include (1) insufficient coverage of Earth science content in general pretraining corpora, (2) the absence of iterative chain-of-thought reasoning modules in standard multimodal fusion backbones, and (3) the scarcity of annotated, multimodal datasets that provide step-by-step scientific rationales.

### 4.3.2 MLLM-BASED METRICS

To further establish the correlation between LLMs and human judgment specifically in the domain of Earth Science VQA, we conducted a human evaluation with four Ph.D. candidates specializing in Earth sciences. They scored a random sample of 160 questions from our MSEarth Open Ended benchmark. The models evaluated included Gemini-2.5-Flash, GPT-4o, InternVL3-78B and QwenVL2.5-72B. Our inter-annotator agreement, measured by Krippendorff's alpha, is 69.5. Following LAVE (Mañas et al., 2024), in order to assess the validity of OE-Eval, we calculated its correlation with human judgment using Spearman's rank correlation coefficients. We derive a single "quality" score from the 4 binary ratings (correct/incorrect) per answer as follows: 1.0 if at least 3 annotators rate the answer as correct, 0.5 if only 2 did so, and 0.0 otherwise. The results of this evaluation are presented in the following table:

| Metric | QwenVL2.5-72B | Gemini-2.5-Flash | GPT-4o | InternVL3-78B | Overall |
|--------|---------------|------------------|--------|---------------|---------|
| BERTScore | 61.16 | 60.34 | 63.06 | 59.79 | 61.09 |
| ROUGE | 67.04 | 66.90 | 70.01 | 63.29 | 66.81 |
| METEOR | 69.00 | 67.12 | 67.34 | 64.55 | 66.75 |
| BLEU | 59.32 | 58.96 | 60.57 | 57.14 | 59.00 |
| OE-Eval | 68.31 | **67.13** | 69.85 | **63.99** | **67.32** |

Table 5: Spearman correlation across models.

From the table's results, OE-Eval demonstrates a higher consistency with human judgment compared to all the considered baselines. Traditional metrics are fundamentally ill-suited for evaluating the scientific nuance and factual correctness required by our benchmark. Our MLLM-based metrics were explicitly designed to address this limitation, and their superiority is empirically proven via alignment with human expert judgments.

### 4.3.3 HUMAN PERFORMANCE BASELINE

To clarify benchmark difficulty and further justify its educational relevance, we also included human expert scores in MSEarth-Bench-mini. Specifically, we hired three human experts, all of whom are Ph.D. students with backgrounds in Earth sciences, to evaluate the tasks. We report their average scores to provide a clear baseline for human performance:

| Model | Atmospheric | Solid Earth Geophysics | Geography | Ecology | Geology | Hydrology | Oceanography | Polar | All |
|-------|-------------|------------------------|-----------|---------|---------|-----------|--------------|-------|-----|
| InternVL3-78B | 50.70% | 29.73% | 28.57% | 47.06% | 25.00% | 51.02% | 25.00% | 30.00% | 47.33% |
| Gemini-2.5-Pro | 46.48% | 58.11% | 35.00% | 35.71% | 47.06% | 50.00% | 59.18% | 50.00% | 51.33% |
| o4-mini | 50.00% | 45.00% | 55.88% | 71.43% | 49.30% | 48.98% | 43.33% | 62.50% | 53.00% |
| Expert | 86.49% | 85.00% | 85.29% | 92.86% | 87.32% | 85.71% | 86.67% | 87.50% | 87.00% |

Table 6: Accuracy on MSEarth-Bench-mini across Earth science domains. Human expert scores are averages of three Ph.D.-level Earth science evaluators.

The results clearly demonstrate that human experts consistently outperform the current MLLMs across all Earth science domains.

## 5 CONCLUSION

In this work, we introduce MSEarth, a graduate-level multimodal dataset designed for MLLMs in geoscientific applications. MSEarth not only serves as a robust test set but also includes rich training resources aimed at enhancing the geoscientific understanding and reasoning capabilities of existing MLLMs. Our evaluation reveals significant gaps in current MLLMs' ability to handle complex, graduate-level geoscientific reasoning, highlighting opportunities for improvement. We believe MSEarth will serve as a valuable resource for advancing MLLMs in scientific reasoning and plan to expand its scope to other scientific domains in future work.

## ETHICS AND REPRODUCIBILITY STATEMENT

All papers used were obtained from OpenDataLab (He et al., 2024b) under the CC BY 4.0 license, which permits adaptation and redistribution with attribution. We strictly adhered to all licensing terms and usage requirements specified by OpenDataLab. This work establishes a benchmark for evaluating the multimodal Earth scientific exploration capabilities of MLLMs in the field of Earth sciences. It has broader positive impacts, including promoting the responsible use of AI in scientific research and enhancing public understanding of Earth sciences. We believe MSEarth will serve as a valuable resource for advancing multimodal language models (MLLMs) in scientific reasoning, and we plan to expand its scope to other scientific domains in future work. The benchmark is publicly available to foster further research and innovation in this field. All data in MSEarth are released anonymously, including the complete dataset on HuggingFace (`https://huggingface.co/MSEarth-Data`) and all data-processing, training, and evaluation code on Anonymous Github (`https://anonymous.4open.science/r/MSEarth-2B3F`)

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

Table 7: Comparison with previous multimodal scientific benchmarks. Task types include OE (Open-ended QA), MCQ (Multiple-choice QA), and CG (Caption Generation).

| Benchmark Dataset | Science Topics | Tasks | Difficulty | Original Questions | Validated |
|---|---|---|---|---|---|
| **Multimodal Scientific Benchmarks** | | | | | |
| ScienceQA (Lu et al., 2022) | General Science | MCQ | Primary | ✗ | ✓ |
| SceMQA (Liang et al., 2024a) | General Science | MCQ,OE | Pre-College | ✗ | ✓ |
| Mmmu (Yue et al., 2024) | General Science | MCQ,OE | College-level | ✗ | ✓ |
| OlympiadBench (He et al., 2024a) | Math, Physics | OE | Competition | ✗ | ✓ |
| EMMA (Hao et al., 2025) | General Science | MCQ | Mixed | ✓ | ✓ |
| **Paper-Based Multimodal Scientific Benchmarks** | | | | | |
| SciFIBench (Roberts et al., 2024) | General Science | MCQ | Graduate-Level | ✗ | ✓ |
| ArxivCap/QA (Li et al., 2024b) | General Science | CG,MCQ | Graduate-Level | ✓ | ✗ |
| MMSci (Li et al., 2024c) | General Science | CG,MCQ | Graduate-Level | ✗ | ✗ |
| MSEarth(Ours) | Earth Science | CG,MCQ,OE | Graduate-Level | ✓ | ✓ |

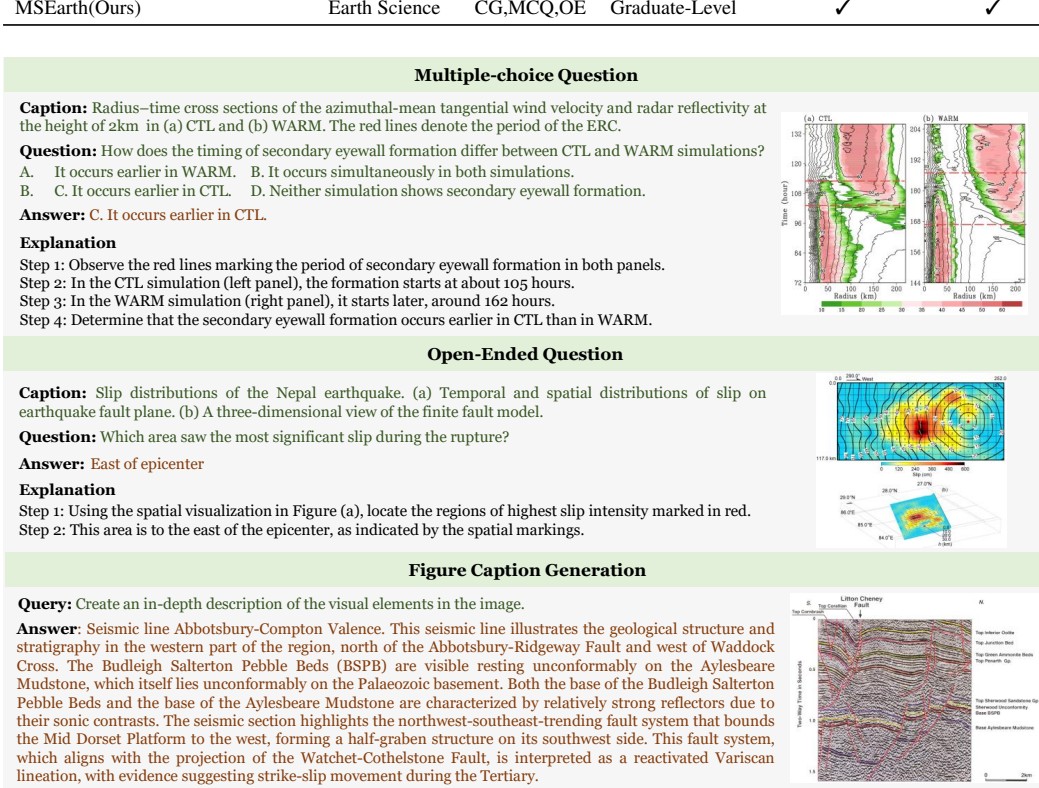

Figure 6: Examples of the three types of scientific question-answering tasks presented in our benchmark.

# A    USAGE OF LANGUAGE MODELS

We utilized a large language model (LLM) to aid in the preparation of this manuscript. Its use was limited to editorial tasks, including proofreading for typographical errors, correcting grammar, and improving the clarity and readability of the text.

# B    BENCHMARK DETAILS

## B.1    FIELD EXPLANATION OF MSEARTH

In Table 8, we provide an explanation of each field for the three tasks in MSEarth.

| Field Name | Input | Description |
|---|:---:|---|
| *Multiple-choice Question* | | |
| question_id | ✗ | The unique identifier for the question. |
| query | ✓ | Contains the original caption, question, and options. |
| response | ✗ | The correct answer to the question. |
| images | ✓ | The file path to the associated image(s). |
| refined_caption | ✗ | The enhanced image description based on the paper content. |
| classification | ✗ | The classification of the question, including its domain and discipline. |
| reasoning_chain | ✗ | The reasoning steps to arrive at the answer. |
| *Open-Ended Question* | | |
| question_id | ✗ | The unique identifier for the question. |
| query | ✓ | Contains the original caption and the question. |
| response | ✗ | The correct answer to the question. |
| images | ✓ | The file path to the associated image(s). |
| refined_caption | ✗ | The enhanced image description based on the paper content. |
| classification | ✗ | The classification of the question, including its domain and discipline. |
| reasoning_chain | ✗ | The reasoning steps to arrive at the answer. |
| *Caption Generation* | | |
| question_id | ✗ | The unique identifier for the question. |
| query | ✓ | The question. |
| response | ✗ | The correct answer (refined caption). |
| images | ✓ | The file path to the associated image(s). |
| context | ✗ | The text from the paper that describes the image. |
| original_caption | ✗ | The original caption of the image. |
| classification | ✗ | The classification of the question, including its domain and discipline. |

Table 8: **Field Descriptions for Different Tasks.** The table provides details about each field, whether it is used as input, and its description. Fields are grouped by task type: MCQ, OE, and Caption Generation.

## B.2 FORMAT CONVERSION

Specifically, our data source is a collection of papers gathered by OpenDataLab (He et al., 2024b) from online resources. These papers were processed using **MinerU**, which converted the textual content of the PDFs into JSON format and saved the images as PNG files. In Figure 7, we present a portion of the content list from a processed PDF paper, highlighting the original caption (raw caption) of the image and the corresponding discussion section. It is evident that the discussion of figures in the paper contains substantial scientific reasoning, which is crucial for a comprehensive understanding of scientific figures.

## B.3 PAPER FILTERING

To classify scientific papers into relevant Earth system categories, we employ a similarity-based approach using pre-trained sentence embeddings and cosine similarity. The process begins by generating embeddings for both the paper's title and predefined keywords using the pre-trained all-MiniLM-L6-v2 (Wang et al., 2020) model, which captures the semantic meaning of textual data. First, we calculate the similarity between the paper's title and a set of general positive keywords, such as "Earth," "Earth system," "hydrosphere," "biosphere," "lithosphere," "atmosphere," and "cryosphere." The cosine similarity is computed between the embedding of the paper's title and the embedding of the general positive keywords. If the similarity score is below a predefined threshold (0.2), the paper is excluded from further analysis, as it is deemed irrelevant to the sciences of the Earth system.

To further filter out irrelevant papers, we calculate the similarity between the paper's title and a set of general negative keywords, such as "cell biology," "virus," "pharmaceuticals," "chemistry," "physics," and "astronomy." If the similarity score exceeds a predefined threshold (0.1), the paper is excluded, as it is likely to belong to unrelated disciplines. For papers that pass the initial filtering, we calculate their similarity with predefined positive classification keywords for each Earth system category (e.g., hydrosphere, biosphere, lithosphere, atmosphere, cryosphere). Each category contains a list of domain-specific keywords. For example, the hydrosphere category includes keywords such

```
Structure of Content List Field

"content_list": [
{
"type": "text",
"text": "LUMINESCENCE STUDIES ON NEOTECTONIC EVENTS IN SOUTH-CENTRAL KUMAUN HIMALAYA -- A FEASIBILITY STUDY",
"text_level": 1
},
........
{
"type": "text",
"text": "The continued northward movement of the Indian plate has caused accumulation of stresses which get periodically released. resulting in
earthquakes and neotectonic activity along faults. Geophysical and structural studies suggest that seismicity in the Himalaya is related to movements along
three major longitudinal thrusts/faults (Fig. 1) viz. the Himalayan Frontal Fault (HFF), the Main Boundary Thrust (MBT) and the Main Central Thrust
(MCT) (Valdiya, 1986, 1988; Nakata. 1989). These thrusts divide the Himalaya into three distinct lithotectonic zones. These zones are further dissected by
numerous transverse faults (Valdiya. 1976; Khattri and Tyagi, 1983a). A concept of locked segments has been propounded to assess the seismogenic
potential of various sectors and is related to accumulation of stresses and their eventual release along the faulted zones (Khattri and Tyagi 1983b). The
repeat frequency of this process of locking of stresses and release of energy is not yet well established due to lack of dating methods. Needless to say, this is
an aspect cardinal to the estimation of seismogenic hazards involved in planning large scale engineering and societal projects in the Himalaya.",
"text_level": 0
},
{
"type": "image",
"img_path": "s3://llm-pipeline-media/pdf-imgs/f5def6d72e7dbfe47ada87d0bc9084997c67f989011db77a255d1817a33fe86b.png",
"img_caption": "FIG. 1. Geological map indicating fault zones and locked segments in Himalaya"
},
........
]
```

Figure 7: Examples of the content list field in a paper.

as "water cycle," "ocean," "rivers," "lakes," and "groundwater," while the biosphere category includes "ecosystem," "biodiversity," "habitat," and "species." The cosine similarity is computed between the paper's title and each keyword within a category, and the average similarity score for each category is calculated.

The category with the highest average similarity score is selected as the most relevant classification for the paper, provided the score exceeds a predefined threshold (0.15). To ensure robustness, we also calculate the similarity between the paper's title and negative classification keywords for each category. For example, the hydrosphere negative keywords include "chemistry," "universe," "planets," and "astronomy," while the biosphere negative keywords include "cell biology," "medicine," and "pharmacology." The final relevance score for each category is computed as the difference between the positive and negative similarity scores, ensuring that papers with high relevance to unrelated fields are excluded. The final classification of a paper is determined by passing the general positive and negative keyword thresholds, identifying the category with the highest positive relevance score adjusted by subtracting the negative relevance score, and ensuring the adjusted relevance score exceeds the classification threshold (0.15). This approach allows us to systematically classify papers into Earth system categories while filtering out irrelevant content, leveraging semantic embeddings and cosine similarity to ensure that the classification is both accurate and interpretable.

### B.4 IMAGE FILTERING

Next, we further filtered the images in these 103,108 papers. Our goal was to retain Earth observation images, as our task focuses on evaluating the model's ability to understand and reason about scientific phenomena in Earth sciences. These images include various types of visual data, such as those representing geophysical processes, atmospheric phenomena, geographic features, weather patterns, and cartographic representations.

To achieve this, we employed a systematic filtering pipeline based on the Qwen2-VL-7B-Instruct model. The filtering process was guided by a carefully designed prompt, which instructed the model to classify each image as either an Earth observation image or not. Specifically, the prompt defined Earth observation images as those depicting remote sensing imagery, atmospheric data visualizations, aerial views of geographical features (e.g., rivers, urban landscapes), weather-related images (e.g., precipitation maps, typhoon tracks), and cartographic representations. Conversely, the prompt explicitly excluded images containing biological entities (e.g., humans, plants, animals), artificial objects (e.g., vehicles, device structures), data visualizations (e.g., statistical charts, line graphs, scatter plots), text-based content, or blank images.

| Category | Type | Keywords |
|---|---|---|
| General | Positive | Earth, Earth system, hydrosphere, biosphere, lithosphere, atmosphere, cryosphere |
| General | Negative | cell biology, virus, pharmaceuticals, chemistry, physics, astronomy, food science, proteins, microbiology |
| Hydrosphere | Positive | water cycle, ocean, rivers, lakes, groundwater, ice caps, aquifers, precipitation, evaporation, humidity |
| Hydrosphere | Negative | chemistry, universe, planets, astronomy, astrophysics, space, stars, galaxy, cosmology |
| Biosphere | Positive | ecosystem, biodiversity, habitat, species, biomes, ecological balance, carbon cycle |
| Biosphere | Negative | cell biology, chemistry, medicine, pharmacology, microbiology, biochemistry, toxicology, pathology, clinical |
| Lithosphere | Positive | earthquake, tectonic plates, earth's crust, minerals, rocks, soil, sediments, mountains, volcanoes, landforms, geological processes |
| Lithosphere | Negative | ancient texts, archaeology, culture, history, artifacts, civilization, prehistoric, mythology, anthropology |
| Atmosphere | Positive | stratosphere, troposphere, weather, climate, greenhouse gases, ozone layer, air pressure, humidity, winds, carbon dioxide, temperature |
| Atmosphere | Negative | universe, galaxy, astronomy, astrophysics, space, stars, planets, cosmology, black holes, nebula, solar system |
| Cryosphere | Positive | glaciers, ice sheets, sea ice, permafrost, snowpack, icebergs, frozen ground, climate change, albedo effect, polar regions |
| Cryosphere | Negative | frozen food, ice cream, refrigeration, freezing, cold storage, ice cubes, food preservation, chilling, frost |

Table 9: Keywords for positive and negative classifications across different Earth system categories. The table includes general keywords as well as specific keywords for hydrosphere, biosphere, lithosphere, atmosphere, and cryosphere.

The filtering process was implemented as follows: for each image, the model was provided with both the image and the prompt, and it generated a binary output ("1" for Earth observation images and "0" otherwise). To ensure robustness, the model's output was validated through multiple sampling attempts with slight variations in generation parameters (e.g., temperature). If the model consistently classified an image as "1," it was retained; otherwise, it was discarded. This iterative and robust classification approach allowed us to minimize false positives and negatives in the filtering process. After this step, we retained around 83K papers, which contained images classified as Earth observation images. These filtered images form the basis for subsequent analysis and evaluation of the model's capabilities in understanding and reasoning about Earth science phenomena.

The following prompt was used to retain Earth observation images:

Analyze the provided image and classify it as an Earth observation image or not.
Earth observation images include, but are not limited to:
- Remote sensing imagery,
- Atmospheric data visualizations,
- Aerial views of geographical features (e.g., rivers, urban landscapes),
- Weather-related images (e.g., precipitation maps, typhoon tracks),
- Cartographic representations.
Exclude images depicting:
- Biological entities (humans, plants, animals),
- Artificial objects (vehicles, device structures),
- Data visualizations (statistical charts, line graphs, scatter plots),
- Text-based content or blank images.
Output format:
- Return "1" if the image is an Earth observation image.

- Return "0" if the image does not meet the Earth observation criteria.

Provide only the numerical output (1 or 0) without any additional text or explanation.

## B.5 CONTENT FILTERING

To construct our benchmark, which requires generating VQA tasks supported by the content of the papers, we ensured that the selected figures not only had captions but were also discussed in detail within the text of the papers. Using regular expressions, we extracted the figure numbers and identified corresponding discussions in the main body of the papers. Figures with discussions exceeding two sentences were included in the final dataset. Finally, we selected 64,560 papers, resulting in a total of 289,891 figures for further processing.

## B.6 PROMPT DESIGNER FOR MSEARTH

The following prompt was used to generate a refined caption:

> You are an expert assistant in scientific image analysis and caption generation. Your task is to rewrite or generate a new, detailed caption for the provided figure using the original caption and only the sentences or information from the Relevant Content that are directly associated with this figure.
> **Please strictly follow these guidelines:**
>   - Assume the figure does not reference or depend on other figures in the document.
>   - Exclude any mention of other figures, their content, or references in the caption.
>   - If subfigures are present, provide specific descriptions for each subfigure accordingly. Otherwise, assume it represents a single figure.
>   - The new caption must be detailed, precise, and include only the relevant details from the provided content.
> **Inputs for caption generation:**
>   - Original Caption: {caption}
>   - Relevant Content: {content}
> Now write a detailed, high-quality caption for this figure below:

The following prompt was used to generate diverse VQAs:

> You are an advanced AI model specialized in generating high-quality Visual Question Answering (VQA) tasks. Your role is to generate a diverse set of VQA questions, answers, and reasoning chains based on the provided visual input (a figure) and its captions.
>
> DEFINITIONS:
>
>   1. **Figure:** A scientific or illustrative figure provided as the primary visual input. Test-takers will analyze this image to answer the questions.
>   2. **Caption:** A concise summary describing key aspects of the Figure.
>   3. **Supplementary:** In-depth information (e.g., summarized expert insight, detailed analysis, or background knowledge) that you can use to assist in designing advanced and meaningful questions. However, test-takers cannot access this information.
>
> INPUT INFORMATION PROVIDED:
>
>   - **Caption:** {raw caption}
>   - **Supplementary:** {refined caption}
>
> TASK INSTRUCTIONS:
>
> Your task is to create a variety of advanced VQA tasks designed to test visual and contextual understanding based on the Figure and Caption. Below are key rules and guidelines you must follow:

1. USE OF INPUT SOURCES:

- Ensure that no question can be answered entirely using Caption without observations from the Figure. Figure content should always serve as a primary source for reasoning.
- **Supplementary Usage:** The correct answers are encouraged to be derived from the Supplementary information. Focus on crafting questions where the Supplementary plays a crucial role in providing the answer.

2. QUESTION TYPES:

- **Multiple Choice Questions (MCQs):** At least **2** questions must be of this type, with 4 distinct options (A-D) and one correct answer.
    - Ensure that only one option can be logically correct based on the provided information (Figure, Caption, and/or Supplementary). Avoid creating options that could lead to ambiguous interpretations or alternate correct answers.
    - Incorrect options must be plausible and relevant to the context but should contain subtle logical flaws or lack supporting evidence when compared to the correct option.
- **Open-Ended Questions:** At least **2** questions must be open-ended, requiring concise and precise answers (no more than 4 words).

3. REASONING CHAINS:

- For every question, you must include a reasoning chain. The chain explains the logical process by which the correct answer can be determined.
- The reasoning chain must:
    - Be clear, step-by-step, and never explicitly mention Caption or Supplementary in `reasoning_chain` (e.g., "According to the Supplementary" or "The Caption states").
    - Use different levels of reasoning complexity.

4. OUTPUT STRUCTURE:

The output must be written in **JSON format** using the structure below:

```
[
    {
        "question_type": MCQ or OE
        "question": "Your question here",
        "options": [A,B,C,D],
        "answer": "Correct option or short answer",
        "reasoning_chain": ["Step 1: ...", ...]
    },
    // Additional questions in the same format...
]
```

5. TASK GUIDELINES:

1. Questions that are grounded in the Supplementary context are highly encouraged. These questions should require the test-taker to refer to in-depth knowledge and insights not immediately visible in the Figure or Caption.
2. Avoid referencing the Supplementary in any question and `reasoning_chain` (e.g., "According to the Supplementary" or "The Supplementary states").

Provide your response below:

## C    MULTI-AGENT VOTING

### C.1    PROMPT

The following prompt was used to generate a normal answer for MCQ:

You are tasked with answering a multiple-choice question about the given input image.

INSTRUCTIONS:

1. Carefully analyze the input image and the provided query.
2. Based on the image, select the correct option (e.g., 'A', 'B', 'C') or directly state the correct option content.
3. Provide reasoning explaining how to derive the correct answer.

INPUT:

- **Query:** {query}

OUTPUT FORMAT:

The output must be written in **JSON format** using the structure below:

```
{
    "answer": "Correct option or short answer",
    "Explanation": "Explaining how to derive correct answer."
}
```

The following prompt was used to generate a enhanced answer for MCQ:

You are tasked with answering a multiple-choice question about the given input image.

INPUT:

- **Question:** {question}
- **Refined Caption:** {caption}

INSTRUCTIONS:

1. Carefully analyze the input image and its caption.
2. Based on the image and caption, select the correct option (e.g., 'A', 'B', 'C') or directly state the correct option content.

OUTPUT FORMAT:

The output must be written in **JSON format** using the structure below:

```
{
    "answer": "Correct option or short answer",
    "Explanation": "Explaining how to derive correct answer."
}
```

## C.2 EXAMPLE OF DIFFERENT LEVELS OF QUESTIONS

Figure 9 illustrates an example of a simple problem in multi-agent voting, while Figure 10 presents an example of a domain-specific problem, and Figure 11 demonstrates an example of a challenging problem. The most notable distinction lies in the varying levels of perceptual ability required by the model: simple and challenging problems primarily differ in the model's ability to perceive and interpret images, whereas domain-specific problems emphasize the model's knowledge in specialized fields. Additionally, the answers to domain-specific questions are often supported by evidence found in the "refined caption" field provided in the paper.

To construct the benchmark datasets, we sampled data from the multi-agent automated filtering process as follows: 900 questions from Phase A, 1800 questions from Phase B, and 300 questions from Phase C were selected to form the multiple-choice question (MCQ) set, while 500 questions from Phase A and 1000 questions from Phase B were selected to form the open-ended question set. All sampled data were subsequently validated by experts to ensure accuracy and quality.

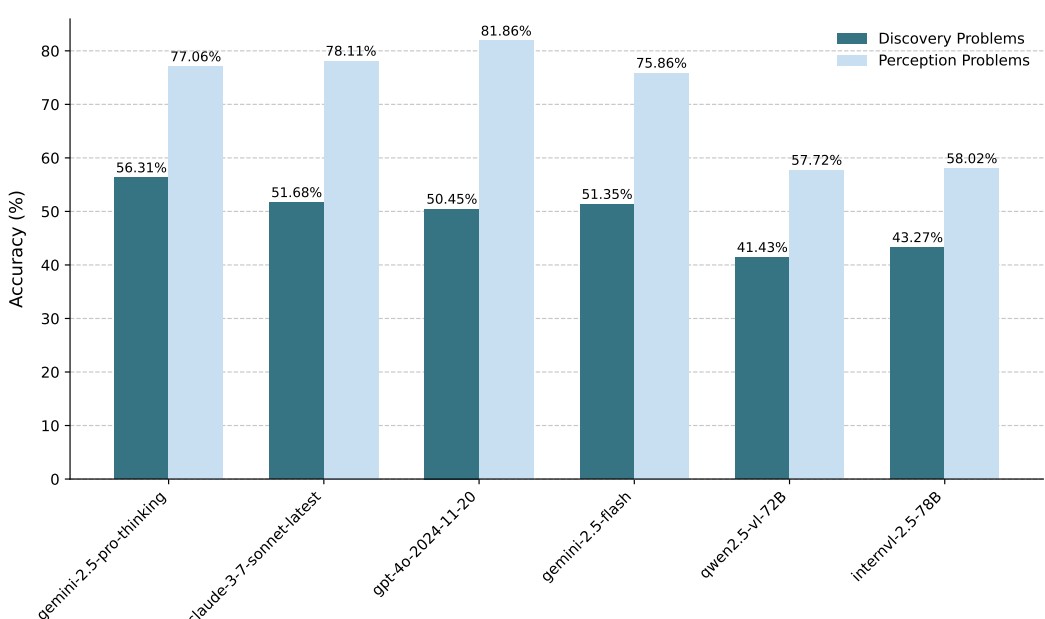

Figure 8: Models' accuracy on reasoning and perception problems.

## D EXPERT VALIDATION

### D.1 DETAILS

We recruited annotators with a background in Earth sciences and a master's degree through an annotation company to label the data. The annotated dataset consists of 3,000 MCQs and 1,500 open-ended QAs. We provided the annotators with figures, queries, reasoning chains, and our processed refined captions to assist them in evaluating whether the provided answers were reasonable. For questions where the answers could not be found in the refined captions, the annotators were required to use their own knowledge to determine the correctness of the answers. If they were unable to make a judgment, such questions were discarded to ensure that the filtered dataset contained only accurate and complete questions. The tasks assigned to the annotators are described below:

The evaluation framework categorizes questions based on several criteria. First, the **Image Type of Reasoning Required** distinguishes between questions involving a single image, where the input consists of just one image, and those with a **Single-image focus**, where multiple images are present but the question pertains to one specific image. Additionally, **Multi-image reasoning** questions require comparing or reasoning across multiple images.

Next, the **Type of Scientific Question** is considered. **Perception Questions** are those where answers can be derived through basic observation, such as identifying position or color. These questions do not have answers in the refined captions and require manual evaluation of their validity. In contrast, **Reasoning Questions** necessitate domain-specific knowledge for answering, and annotators must verify if the answer can be derived from the refined caption field.

The **Completeness of Questions** is another criterion, where questions are classified as **Complete** if all necessary information is provided in the question or image, and **Incomplete** if missing information makes it difficult or impossible to answer.

Finally, the **Correctness of Questions** assesses whether the provided answer is accurate, categorizing them as **Correct** or **Incorrect** based on the accuracy of the answer.

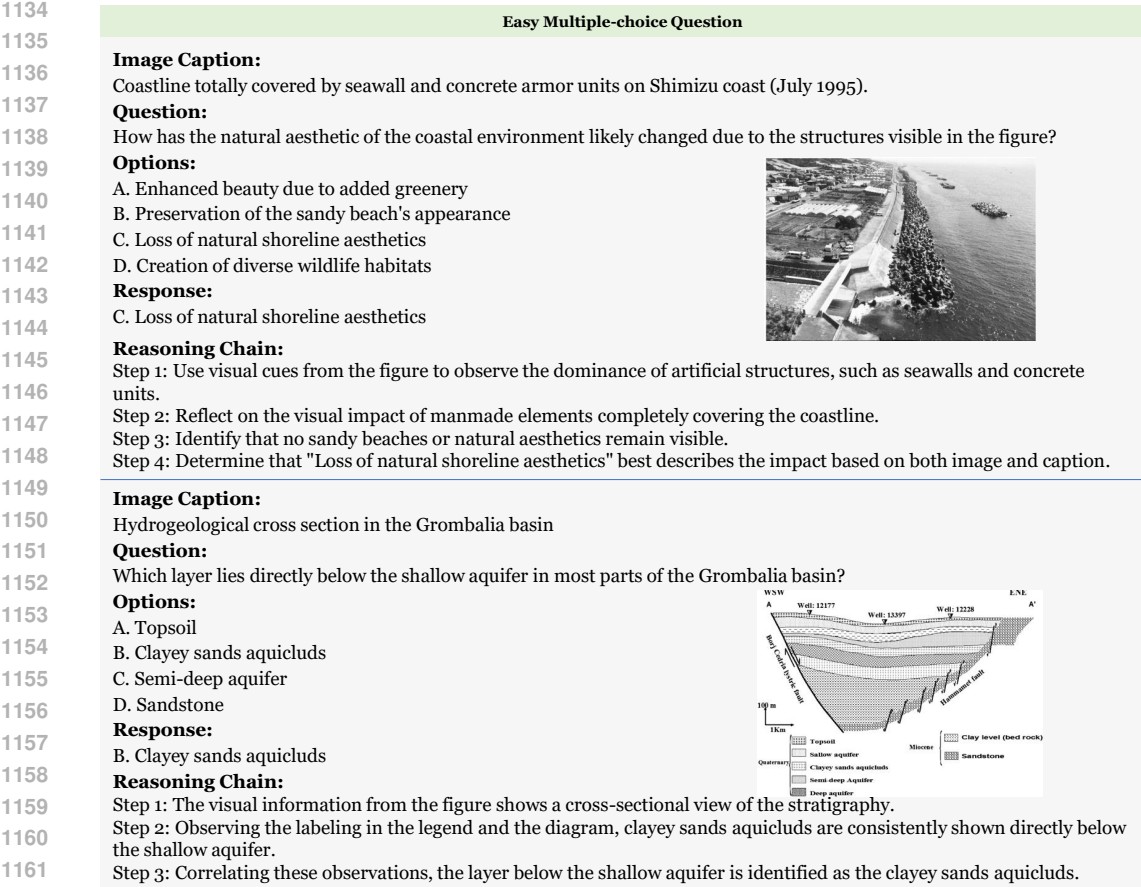

**Easy Multiple-choice Question**

**Image Caption:**
Coastline totally covered by seawall and concrete armor units on Shimizu coast (July 1995).

**Question:**
How has the natural aesthetic of the coastal environment likely changed due to the structures visible in the figure?

**Options:**
A. Enhanced beauty due to added greenery
B. Preservation of the sandy beach's appearance
C. Loss of natural shoreline aesthetics
D. Creation of diverse wildlife habitats

**Response:**
C. Loss of natural shoreline aesthetics

**Reasoning Chain:**
Step 1: Use visual cues from the figure to observe the dominance of artificial structures, such as seawalls and concrete units.
Step 2: Reflect on the visual impact of manmade elements completely covering the coastline.
Step 3: Identify that no sandy beaches or natural aesthetics remain visible.
Step 4: Determine that "Loss of natural shoreline aesthetics" best describes the impact based on both image and caption.

**Image Caption:**
Hydrogeological cross section in the Grombalia basin

**Question:**
Which layer lies directly below the shallow aquifer in most parts of the Grombalia basin?

**Options:**
A. Topsoil
B. Clayey sands aquicluds
C. Semi-deep aquifer
D. Sandstone

**Response:**
B. Clayey sands aquicluds

**Reasoning Chain:**
Step 1: The visual information from the figure shows a cross-sectional view of the stratigraphy.
Step 2: Observing the labeling in the legend and the diagram, clayey sands aquicluds are consistently shown directly below the shallow aquifer.
Step 3: Correlating these observations, the layer below the shallow aquifer is identified as the clayey sands aquicluds.

Figure 9: An example of easy multiple-choice VQA.

After manual screening, 216 invalid entries were identified in the MCQ task, and 89 invalid entries were found in the open-ended task. To evaluate the effectiveness of our multi-agent filtering process, we conducted a statistical analysis of the three phases of data. In Phase A, 59 out of 900 sampled questions were deemed invalid after manual review; in Phase B, 80 out of 1800 questions were invalid; and in Phase C, 77 out of 300 questions were invalid. These results demonstrate the utility of the initial model-based filtering: questions supported by refined captions and correctly answered by most models (Phases A and B) tend to be of higher quality, while questions filtered in Phase C exhibit lower quality.

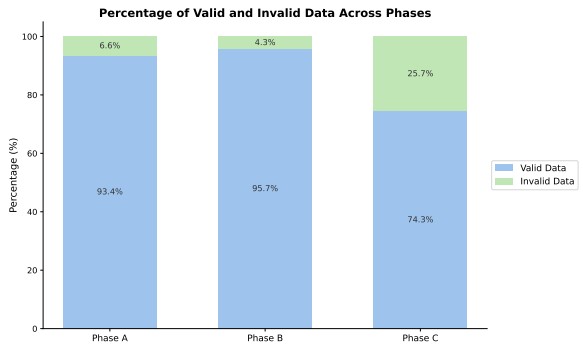

Figure 12: Proportion of valid and invalid data after manual screening across different phases. Phase B, or Specialized VQA, exhibits the highest quality.

**Moderate Multiple-choice Question**

**Image Caption:**
Change in sea level pressure between different pairs of sensitivity experiments with the sea surface temperature gradient altered in the Atlantic Ocean, Pacific Ocean, or in combinations in both oceans.
....
Bottom Panel: Difference between experiments AdPi and AiPd. Most changes outside of the range shown ($-2.2$ to $2.2~\mathrm{mb}$) are significant at the 95% level.

**Question:**
What mechanism best explains the Southern Hemisphere pressure changes observed in the bottom panel (AdPi-AiPd)?

**Options:**
A. Pacific tropical warming amplifies wave refraction.
B. High-latitude cooling destabilizes atmospheric conditions.
C. Sea ice variations influence atmospheric circulation.
D. Increased sensible heat transports planetary momentum.

**Response:**
C. Sea ice variations influence atmospheric circulation.

**Reasoning Chain:**
Step 1: Look at the bottom panel (AdPi-AiPd), which demonstrates significant pressure responses in the Southern Hemisphere.
Step 2: Identify pressure anomalies during Southern Hemisphere summer that align with locations of seasonal sea ice.
Step 3: Use context from the caption and figure dynamics to deduce influence from sea ice variations.

**Refined Caption:**
Sea level pressure differences between sensitivity experiments with altered sea surface temperature (SST) gradients in the Atlantic and Pacific Oceans:

.....
•Bottom Panel: The difference between Experiment AdPi (Atlantic high-latitude warming and tropical cooling combined with Pacific tropical warming and high-latitude cooling) and Experiment AiPd (the reverse gradient alterations). This configuration produces a large positive NAO change, with warm tropical Pacific SSTs driving equatorward wave refraction and poleward angular momentum transport, while warm northern North Atlantic SSTs reduce low-altitude northward sensible heat transport and destabilize the local atmosphere. Notable responses are also observed in the Southern Hemisphere, even during summer, which are linked to sea ice variations. Most changes outside the range of $-2.2$ to $2.2~\mathrm{mb}$ are statistically significant at the 95% confidence level.

Figure 10: An example of specialized multiple-choice VQA.

**Hard Multiple-choice Question**

**Image Caption:**
MERIS RGB image of scene over Mediterranean Sea from 2 May 2003 (left) and zoom into daily composite of TCWV (middle) and uncertainty of TCWV (right) from MERIS and SSM/I measurements.

**Question:**
In which regions is TCWV density the highest based on the middle panel?

**Options:**
A. Over the Mediterranean Sea
B. Over land near coastal areas
C. Over desert areas in the bottom left
D. In the southernmost water regions

**Response:**
D. In the southernmost water regions

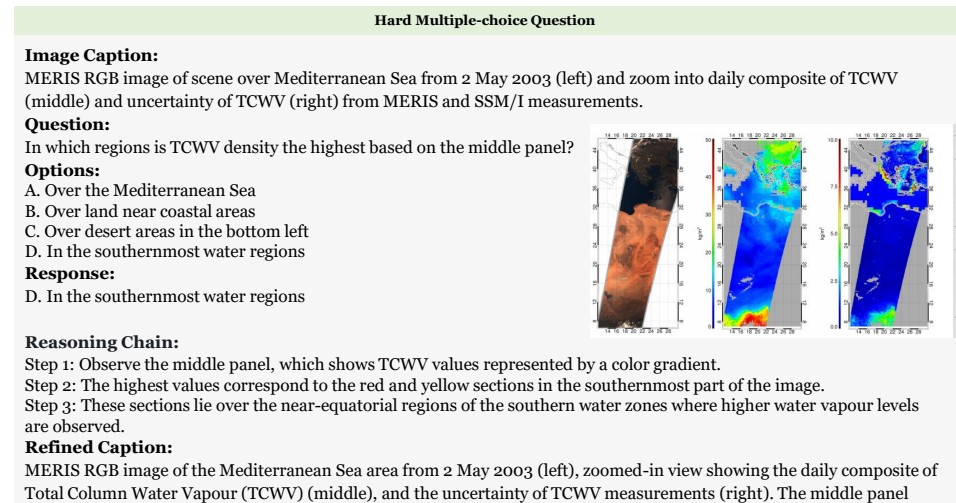

**Reasoning Chain:**
Step 1: Observe the middle panel, which shows TCWV values represented by a color gradient.
Step 2: The highest values correspond to the red and yellow sections in the southernmost part of the image.
Step 3: These sections lie over the near-equatorial regions of the southern water zones where higher water vapour levels are observed.

**Refined Caption:**
MERIS RGB image of the Mediterranean Sea area from 2 May 2003 (left), zoomed-in view showing the daily composite of Total Column Water Vapour (TCWV) (middle), and the uncertainty of TCWV measurements (right). The middle panel illustrates the smooth transition of the water vapour field between land and ocean, with increased uncertainty in coastal areas. This uncertainty is linked to the use of MERIS data to fill gaps in the SSM/I measurements. The right panel highlights the regions of elevated uncertainty, particularly along the coast. Additionally, the figure emphasizes MERIS's high sensitivity to small-scale variations in the water vapour field, as seen over Western Turkey.

Figure 11: An example of hard multiple-choice VQA.

# E CLASSIFICATION OF RESEARCH PROBLEMS IN EARTH SCIENCES

Under the framework of the five major spheres, we further categorized the generated research problems into specific academic disciplines according to a standardized classification system. Within

Table 10: Main statistics in MSEarth-Bench.

| Statistic | Number |
|---|---|
| Total questions | 7,195 |
| MCQ | 2,784 |
| Questions with single images | 1,255 (45.1%) |
| Questions with multiple images | 1,529 (54.9%) |
|   * Single-image focus | ≈1,164 (41.8%) |
|   * Multi-image relational | ≈365 (13.1%) |
| Reasoning Question | 2,117 (76.0%) |
| Perception Question | 667 (24.0%) |
| Open-Ended | 1,411 |
| Questions with single images | 679 (45.1%) |
| Questions with multiple images | 832 (54.9%) |
|   * Single-image focus | ≈619 (41.8%) |
|   * Multi-image relational | ≈113 (13.1%) |
| Captioning | 3,000 |
| Average caption length | 37.71 |
| Average refined caption length | 137.47 |

the broad category of Earth Sciences, we refined the classification into detailed sub-disciplines or sub-fields. The classification process involves three main steps: first, identifying the primary sphere to which the research problem belongs, selecting from eight major disciplines (referred to as primary spheres), including Atmospheric Sciences, Ecology and Biosciences, Hydrology, Oceanography, Geology, Geography, Solid Earth Geophysics, and Polar Science. Second, the classification is further refined by selecting the most appropriate sub-discipline or sub-field from a detailed hierarchy. Third, for interdisciplinary problems, the primary classification is clearly stated, and any relevant secondary classifications are noted. This hierarchical approach ensures a systematic and precise categorization of research problems, enabling a deeper understanding of their academic and scientific context.

**Summary of Classification:** The classification system includes a total of 8 first-level disciplines and 66 second-level disciplines. Each research problem is assigned to one of the primary disciplines and further refined into a specific sub-discipline based on its characteristics and context.

---

You are tasked with classifying a research problem into one of the Earth's spheres and refining it into a specific sub-discipline or sub-field.

INSTRUCTIONS:

1. Carefully analyze the input, which includes the research question, paper title, and any additional information derived from images (e.g., visual data descriptions or extracted features).
2. Identify the primary sphere (**Atmospheric Sciences, Ecology and Biosciences, Hydrology, Oceanography, Geology, Geography, Solid Earth Geophysics, or Polar Science**) that the problem belongs to.
3. Refine the classification by selecting the most appropriate sub-discipline or sub-field from the hierarchy.
4. If the problem spans multiple spheres or disciplines, clearly state the primary classification and mention any relevant secondary classifications.

CLASSIFICATION HIERARCHY:

**Atmospheric Sciences:** Atmospheric Chemistry, Meteorology, Climatology, Hydrometeorology, Paleoclimatology, Atmospheric Physics, Numerical Weather Prediction and Simulation, Atmospheric Remote Sensing.
**Ecology and Biosciences:** Regional Ecology, Population Ecology, Community Ecology, Ecosystem Ecology, Ecological Engineering, Restoration Ecology, Landscape Ecology, Aquatic Ecology and Limnological Ecoly, Biogeochemistry, Biogeography.

**Hydrology:** Hydrology, Hydrogeology, Limnology, River Hydrology and Estuarine Hydrology, Groundwater Hydrology, Regional Hydrology, Ecohydrology, Hydrological Physics, Hydrological Geography, Hydrological Meteorology, Hydrological Measurement, Hydrological Cartography.

**Oceanography:** Ocean Chemistry, Ocean Physics, Ocean Biology, Ocean Geology, Remote Sensing Oceanography, Environmental Oceanography, Marine Resources Science.

**Geology:** Economic Geology, Engineering Geology, Environmental Geology, Quaternary Geology, Sedimentology, Stratigraphy, Paleogeography, Volcanology, Mineralogy and Petrology, Regional Geology, Remote Sensing Geology.

**Geography:** Physical Geography, Human Geography, Regional Geography, Urban Geography, Tourism Geography, World Geography, Historical Geography, Geomorphology, Biogeography, Chemical Geography, Other Disciplines in Geography.

**Solid Earth Geophysics:** Geodynamics, Seismology, Geomagnetism, Gravimetry, Geoelectricity, Geothermal Science, Tectonophysics, Exploration Geophysics, Computational Geophysics, Experimental Geophysics, Other Disciplines in Solid Earth Geophysics.

**Polar Science:** Polar Ecology, Polar Oceanography, Glaciology, Permafrost Science, Polar Climate Science.

INPUT:

- **Paper Title:** {paper_title}
- **Research Question:** {research_question}
- **Image Information:** {image_caption}

OUTPUT FORMAT:

The output must be written in **JSON format** using the structure below:

```
{
    "primary_sphere": ,
    "primary_sub_discipline": ,
    "secondary_sphere": "Ecology and Biosciences",
    "secondary_sub_discipline": "Aquatic Ecology"
}
```

Table 11: **Top Sub-disciplines in Various Scientific Subjects.** The table lists the top three sub-disciplines by count within each major scientific subject.

| Subject | Top 1 Sub-subject | | Top 2 Sub-subject | | Top 3 Sub-subject | |
|---|---|---|---|---|---|---|
| Hydrology | River Hydrology and Estuarine Hydrology | 805 | Groundwater Hydrology | 790 | Limnology | 439 |
| Ecology and Biosciences | Aquatic Ecology and Limnological Ecology | 562 | Landscape Ecology | 298 | Ecosystem Ecology | 280 |
| Geology | Sedimentology | 1068 | Quaternary Geology | 298 | Structural Geology | 215 |
| Solid Earth Geophysics | Seismology | 845 | Tectonophysics | 343 | Exploration Geophysics | 74 |
| Geography | Physical Geography | 1575 | Urban Geography | 76 | Geomorphology | 40 |
| Polar Science | Glaciology | 352 | Polar Climate Science | 46 | Permafrost Science | 31 |
| Atmospheric Sciences | Meteorology | 920 | Climatology | 619 | Atmospheric Remote Sensing | 159 |
| Oceanography | Ocean Physics | 698 | Ocean Geology | 163 | Environmental Oceanography | 104 |

# F  MLLMs' VERSIONS

For open-source models, we use vllm (Kwon et al., 2023) for local testing; for proprietary models, we conduct tests via API calls. The download paths for specific models and the versions of models accessed via API are provided in Figure 12.

# G  EVALUATION METRICS

## G.1  MLLM-BASED METRICS

Following G-Eval (Liu et al., 2023b), we utilize MLLM (Qwen2.5-VL-72B) with a specialized prompt to compute a factual scientific score. For the captioning task, we define a Cap-Eval score

Table 12: Evaluated MLLMs in our experiments with their versions or Huggingface model paths.

| Open-source Models | |
|---|---|
| **Model** | **Model path** |
| Qwen2.5-VL-7B | `https://huggingface.co/Qwen/Qwen2.5-VL-7B-Instruct` |
| Qwen2.5-VL-32B | `https://huggingface.co/Qwen/Qwen2.5-VL-32B-Instruct` |
| Qwen2.5-VL-72B | `https://huggingface.co/Qwen/Qwen2.5-VL-72B-Instruct` |
| InternVL2.5-8B | `https://huggingface.co/OpenGVLab/InternVL2_5-8B` |
| InternVL2.5-78B | `https://huggingface.co/OpenGVLab/InternVL2_5-78B` |
| InternVL3-78B | `https://huggingface.co/OpenGVLab/InternVL3-78B` |
| LLaVA-onvision-72B | `https://huggingface.co/llava-hf/llava-onevision-qwen2-72b-ov-hf` |
| Llama3.2-90B-Vision | `https://huggingface.co/meta-llama/Llama-3.2-90B-Vision` |
| DeepSeek-VL2 | `https://huggingface.co/deepseek-ai/deepseek-vl2` |
| Intern-S1-mini | `https://huggingface.co/internlm/Intern-S1-mini` |
| Proprietary Models | |
| **Model** | **Model versioning** |
| GPT-4o | `GPT-4o-20` |
| Gemini-2.5-Pro-Thinking | `gemini-2.5-pro-preview-05-06` |
| Gemini-2.5-Flash | `Gemini-2.5-Flash-preview-04-17` |
| Gemini-2.5-Flash-Thinking | `Gemini-2.5-Flash-preview-04-17` |
| Claude-3.7-Sonnet | `Claude-3.7-Sonnet-20250219` |
| Claude-3.5-Haiku | `claude-3-5-haiku-20241022` |
| GPT-4o-mini | `GPT-4o-mini-2024-07-18` |

ranging from 1 to 5, where higher scores indicate better caption quality. For the open-ended QA task, we introduce OE-Eval, which evaluates the reasonableness of generated answers using a binary 0/1 scoring system.

The following prompt was used for Cap-Eval:

---

Evaluate the quality of a generated caption for a geoscience research paper figure or image.

EVALUATION CRITERIA:

1. **Scientific Accuracy:** Does the generated caption accurately describe the scientific content of the figure or image?
2. **Clarity and Coherence:** Is the caption well-structured, logically organized, and easy to understand?
3. **Relevance and Completeness:** Does the caption provide all necessary information to understand the figure or image?

EVALUATION STEPS:

1. Compare the **Generated Caption** to the **Standard Caption**. Assess whether the generated caption aligns with the scientific content and intent of the standard caption.
2. Assign a score for coherence on a scale of 1 to 5, where 1 is the lowest and 5 is the highest, based on the Evaluation Criteria.

INPUT:

- **Standard Caption:** {response}
- **Generated Caption:** {generated_caption}

IMPORTANT INSTRUCTIONS:

- Only output the score in the specified JSON format.
- Do not provide any explanations, comments, or additional text.

OUTPUT FORMAT:

The output must be written in **JSON format** using the structure below:

---

```
{
    "score": 1-5
}
```

The following prompt was used for OE-Eval:

You are tasked with evaluating the correctness of a generated answer to an open-ended question about a given input image.

INPUT:

- **Question:** {query}
- **Refined Caption:** {refined caption}
- **Standard Answer:** {response}
- **Generated Answer:** {generated_answer}

INSTRUCTIONS:

1. Based on the refined caption, question, and standard answer, determine if the generated answer is correct.
2. Only output the determination in the specified JSON format.
3. Do not provide any explanations, comments, or additional text.

OUTPUT FORMAT:

The output must be written in **JSON format** using the structure below:

```
{
    "is_correct": true or false
}
```

To further establish the correlation between LLMs and human judgment specifically in the domain of Earth Science VQA, we conducted a human evaluation with four Ph.D. candidates specializing in Earth sciences. They scored a random sample of 160 questions from our MSEarth Open Ended benchmark. The models evaluated included Gemini-2.5-Flash, GPT-4o, InternVL3-78B and QwenVL2.5-72B. Our inter-annotator agreement, measured by Krippendorff's alpha, is 69.5. Following LAVE (Mañas et al., 2024), in order to assess the validity of OE-Eval, we calculated its correlation with human judgment using Spearman's rank correlation coefficients. We derive a single "quality" score from the 4 binary ratings (correct/incorrect) per answer as follows: 1.0 if at least 3 annotators rate the answer as correct, 0.5 if only 2 did so, and 0.0 otherwise. The results of this evaluation are presented in the following table:

| Metric | QwenVL2.5-72B | Gemini-2.5-Flash | GPT-4o | InternVL3-78B | Overall |
|--------|---------------|------------------|--------|---------------|---------|
| BERTScore | 61.16 | 60.34 | 63.06 | 59.79 | 61.09 |
| ROUGE | 67.04 | 66.90 | 70.01 | 63.29 | 66.81 |
| METEOR | 69.00 | 67.12 | 67.34 | 64.55 | 66.75 |
| BLEU | 59.32 | 58.96 | 60.57 | 57.14 | 59.00 |
| OE-Eval | 68.31 | **67.13** | 69.85 | **63.99** | **67.32** |

Table 13: Spearman correlation across models.

From the table's results, OE-Eval demonstrates a higher consistency with human judgment compared to all the considered baselines.

### G.2 SIMILARITY-BASED METRICS

In cases where some models fail to strictly follow instructions and only output the correct answer, resulting in regular expression matching failures, we use the all-MiniLM-L6-v2 model (Wang et al.,

2020) to calculate the similarity between the model's output and each option. The option with the highest similarity is then selected as the model's answer.

# H DETAILED MSEARTH-MCQ RESULTS

The inputs to our model, MSEarth, consist of images, questions, and the original captions. The original captions provide contextual information about the images, such as the meanings of specific symbols. Therefore, we conducted tests on different models to evaluate their performance with and without the original captions.

Table 14: Accuracies (%) of different models on multiple-choice questions. The best results are highlighted in bold, with the second-best underlined. OC: original caption.

| Model | Input OC | Image-Type SINGLE | MULTI | CROSS | Task Type REASONING | PERCEPT | Overall ACC |
|---|---|---|---|---|---|---|---|
| *Open-source Models* | | | | | | | |
| LLaVA-onvision-72B | ✗ | 49.40 | 45.52 | 41.10 | 41.92 | 61.86 | 46.69 |
| Qwen2.5-VL-7B | ✗ | 39.12 | 35.65 | 39.18 | 37.27 | 38.98 | 37.68 |
| Qwen2.5-VL-32B | ✗ | 42.07 | 39.78 | 40.00 | 37.03 | 52.92 | 40.84 |
| Qwen2.5-VL-72B | ✗ | 47.65 | 43.30 | 43.84 | 41.43 | 57.72 | 45.33 |
| InternVL2-8B | ✗ | 35.94 | 34.11 | 32.33 | 34.25 | 36.13 | 34.70 |
| InternVL2.5-78B | ✗ | 48.13 | 45.88 | 45.21 | 43.27 | 58.02 | 46.80 |
| InternVL3-78B | ✗ | 51.95 | 44.85 | 45.75 | 44.54 | 59.67 | 48.17 |
| Llama3.2-90B-Vision | ✗ | 44.30 | 40.64 | 36.16 | 38.64 | 51.42 | 41.70 |
| DeepSeek-VL2 | ✗ | 45.42 | 42.70 | 46.85 | 43.74 | 46.78 | 44.47 |
| LLaVA-onvision-72B | ✓ | 53.55 | 49.48 | 47.95 | 46.58 | 65.52 | 51.11 |
| Qwen2.5-VL-7B | ✓ | 47.65 | 44.07 | 37.53 | 40.53 | 58.47 | 44.83 |
| Qwen2.5-VL-32B | ✓ | 52.59 | 46.99 | 43.84 | 42.47 | 70.16 | 49.10 |
| Qwen2.5-VL-72B | ✓ | 52.11 | 50.43 | 46.30 | 44.40 | 70.46 | 50.65 |
| InternVL2-8B | ✓ | 44.86 | 43.99 | 38.36 | 38.97 | 58.47 | 43.64 |
| InternVL2.5-78B | ✓ | 53.23 | 49.74 | 44.38 | 43.17 | 74.21 | 50.61 |
| InternVL3-78B | ✓ | 57.53 | 51.37 | 45.48 | 47.00 | 73.61 | 53.38 |
| Llama3.2-90B-Vision | ✓ | 45.98 | 40.46 | 38.90 | 38.26 | 56.97 | 42.74 |
| DeepSeek-VL2 | ✓ | 52.43 | 49.23 | 44.66 | 46.06 | 62.82 | 50.07 |
| *Proprietary Models* | | | | | | | |
| Gemini-2.5-Flash | ✓ | 58.33 | 54.55 | 53.42 | 49.98 | 75.56 | 56.11 |
| Gemini-2.5-Flash-Thinking | ✓ | 60.64 | 54.64 | 53.70 | 51.35 | 75.86 | 57.22 |
| Gemini-2.5-Pro-Thinking | ✓ | **64.78** | **59.36** | 55.34 | **56.31** | 77.06 | **61.28** |
| Claude-3.5-Haiku | ✓ | 49.48 | 47.16 | 42.47 | 42.18 | 64.77 | 47.59 |
| Claude-3.7-Sonnet | ✓ | 59.52 | 56.53 | **57.53** | 51.68 | 78.11 | 58.01 |
| GPT-4o-mini | ✓ | 52.51 | 48.63 | 43.01 | 43.65 | 68.67 | 49.64 |
| GPT-4o | ✓ | 63.03 | 55.76 | 47.67 | 50.45 | **81.86** | 57.97 |

For open source models, we performed experiments ino settings: with and without the original caption. The results show that providing the original caption improves performance in all tasks. Notably, the improvement is more significant for perception tasks compared to reasoning tasks. This is likely because perception tasks rely more heavily on understanding the image content, and the original caption provides helpful contextual information for interpreting the image.

We have compiled several case studies to illustrate the necessity of the original caption when answering questions in certain situations. In example 14, if the original caption is not provided, InternVL3-78B will be unable to accurately determine that the geographical location is in Germany, resulting in an incorrect answer. In contrast, some proprietary models may possess stronger perceptual capabilities and can correctly identify the location as Germany even without the original caption. Similarly, in example 15, providing the original caption aids the model in understanding the image, thereby facilitating task completion. Both scenarios are prevalent in scientific question-answering contexts. To address this, we conducted separate experiments and explicitly integrated these settings into the design of the MSEarth-MCQ task.

## I    RESULTS WITH COMPUTE SCALING

From the main experiments, it is evident that the performance of various models declines significantly on questions requiring specialized knowledge. To explore whether existing methods can enhance model performance on such questions, we sampled 300 specialized questions from the MCQ dataset to create the MSEarth-mini set. We then evaluated the effectiveness of Chain-of-Thought (CoT) reasoning and majority voting mechanism, which selects the most frequent response among (N) candidate responses; in the case of a tie, one of the most frequent answers is randomly chosen. The results are presented in figure 13. Notably, for the Gemini-Pro-thinking model, which inherently incorporates a thinking mechanism, introducing CoT reasoning led to a decline in performance. Similarly, for some open-source models, such as Qwen and InternVL, the addition of CoT reasoning also resulted in performance degradation. However, the majority voting mechanism proved effective for most models.

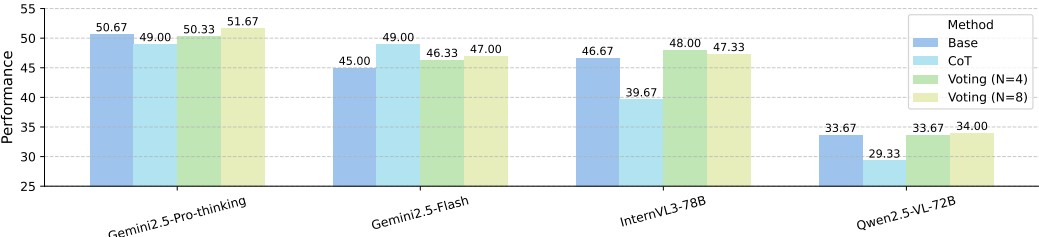

Figure 13: Performance of different strategies on MSEarth-mini.

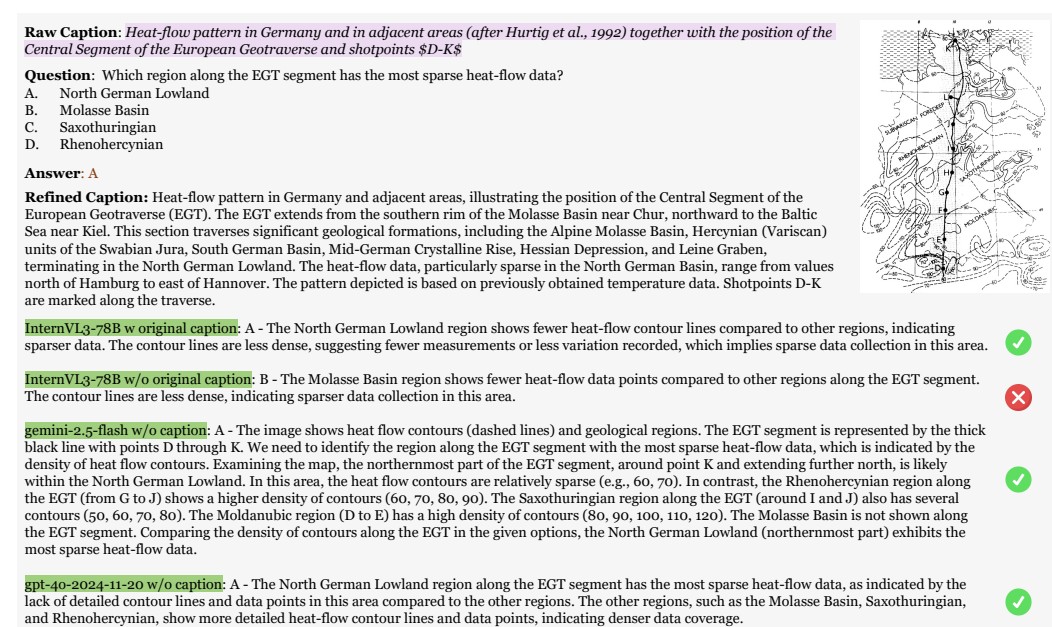

Figure 14: Performance comparison of different models under two settings: with and without the original caption. The results indicate that more powerful models exhibit less dependence on the original caption, highlighting their ability to interpret image content independently.

## J    HUMAN PERFORMANCE BASELINE

To clarify benchmark difficulty and further justify its educational relevance, we also included human expert scores in MSEarth-Bench-mini. Specifically, we hired three human experts, all of whom are Ph.D. students with backgrounds in Earth sciences, to evaluate the tasks. We report their average scores to provide a clear baseline for human performance:

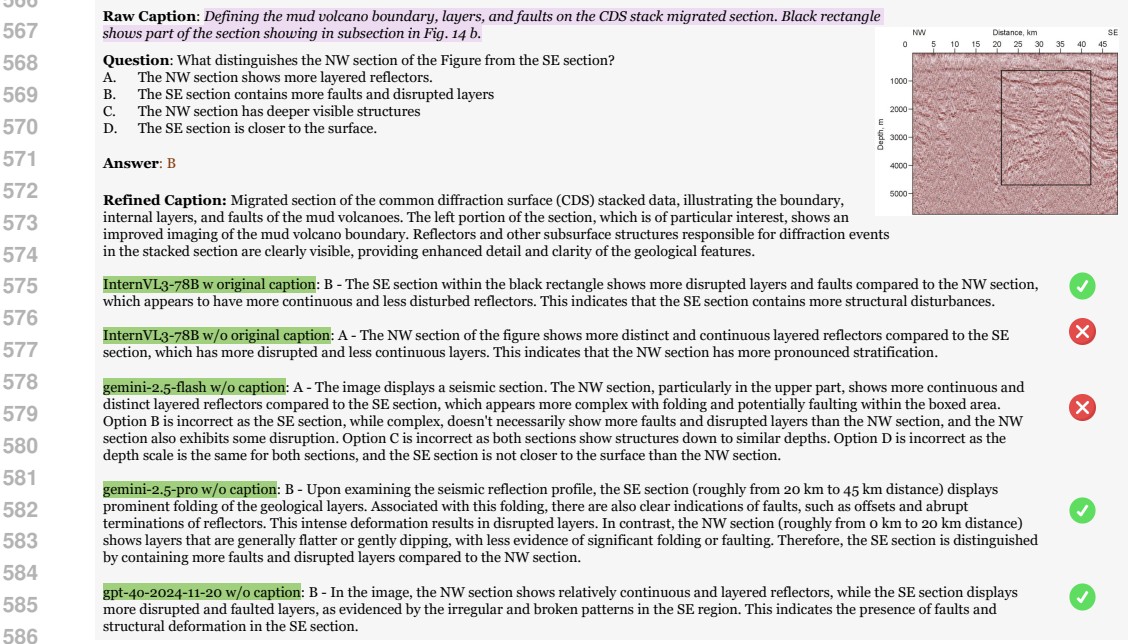

**Raw Caption**: *Defining the mud volcano boundary, layers, and faults on the CDS stack migrated section. Black rectangle shows part of the section showing in subsection in Fig. 14 b.*

**Question**: What distinguishes the NW section of the Figure from the SE section?
A.  The NW section shows more layered reflectors.
B.  The SE section contains more faults and disrupted layers
C.  The NW section has deeper visible structures
D.  The SE section is closer to the surface.

**Answer**: B

**Refined Caption:** Migrated section of the common diffraction surface (CDS) stacked data, illustrating the boundary, internal layers, and faults of the mud volcanoes. The left portion of the section, which is of particular interest, shows an improved imaging of the mud volcano boundary. Reflectors and other subsurface structures responsible for diffraction events in the stacked section are clearly visible, providing enhanced detail and clarity of the geological features.

InternVL3-78B w original caption: B - The SE section within the black rectangle shows more disrupted layers and faults compared to the NW section, which appears to have more continuous and less disturbed reflectors. This indicates that the SE section contains more structural disturbances.

InternVL3-78B w/o original caption: A - The NW section of the figure shows more distinct and continuous layered reflectors compared to the SE section, which has more disrupted and less continuous layers. This indicates that the NW section has more pronounced stratification.

gemini-2.5-flash w/o caption: A - The image displays a seismic section. The NW section, particularly in the upper part, shows more continuous and distinct layered reflectors compared to the SE section, which appears more complex with folding and potentially faulting within the boxed area. Option B is incorrect as the SE section, while complex, doesn't necessarily show more faults and disrupted layers than the NW section, and the NW section also exhibits some disruption. Option C is incorrect as both sections show structures down to similar depths. Option D is incorrect as the depth scale is the same for both sections, and the SE section is not closer to the surface than the NW section.

gemini-2.5-pro w/o caption: B - Upon examining the seismic reflection profile, the SE section (roughly from 20 km to 45 km distance) displays prominent folding of the geological layers. Associated with this folding, there are also clear indications of faults, such as offsets and abrupt terminations of reflectors. This intense deformation results in disrupted layers. In contrast, the NW section (roughly from 0 km to 20 km distance) shows layers that are generally flatter or gently dipping, with less evidence of significant folding or faulting. Therefore, the SE section is distinguished by containing more faults and disrupted layers compared to the NW section.

gpt-4o-2024-11-20 w/o caption: B - In the image, the NW section shows relatively continuous and layered reflectors, while the SE section displays more disrupted and faulted layers, as evidenced by the irregular and broken patterns in the SE region. This indicates the presence of faults and structural deformation in the SE section.

Figure 15: Performance comparison of different models under two settings: with and without the original caption.

| Model | Atmospheric | Solid Earth Geophysics | Geography | Ecology | Geology | Hydrology | Oceanography | Polar | All |
|---|---|---|---|---|---|---|---|---|---|
| InternVL3-78B | 50.70% | 29.73% | 28.57% | 47.06% | 25.00% | 51.02% | 25.00% | 30.00% | 47.33% |
| gemini-2.5-pro | 46.48% | 58.11% | 35.00% | 35.71% | 47.06% | 50.00% | 59.18% | 50.00% | 51.33% |
| o4-mini | 50.00% | 45.00% | 55.88% | 71.43% | 49.30% | 48.98% | 43.33% | 62.50% | 53.00% |
| Expert | 86.49% | 85.00% | 85.29% | 92.86% | 87.32% | 85.71% | 86.67% | 87.50% | 87.00% |

Table 15: Accuracy on MSEarth-Bench-mini across Earth science domains. Human expert scores are averages of three Ph.D.-level Earth science evaluators.

The results clearly demonstrate that human experts consistently outperform the current MLLMs across all Earth science domains.

## K    IMPACT OF EXPLICIT REASONING

We assess the impact of explicit chain-of-thought (CoT) prompting on MSEarth-Bench-mini across both open-source and proprietary LVLMs. For open-source models, we compare InternVL3 and QwenVL2.5; for proprietary models, we examine the Gemini-2.5-Flash series. Within the proprietary family, variants with dedicated "thinking" capabilities (e.g., Gemini-2.5-Flash-Thinking) generally outperform counterparts without such capabilities (e.g., Gemini-2.5-Flash). In contrast, for open-source models, adding explicit CoT sometimes leads to performance declines, which we hypothesize stems from limited training for explicit reasoning behaviors (e.g., GRPO-style preference optimization).

To further probe the role of explicit CoT, we include GPT-o4-mini, which exposes configuration options that control reasoning depth (low, medium, high), roughly corresponding to the length of the reasoning chain. Results are shown in Table 16.

Overall, we observe the following:

Models explicitly equipped and trained for "thinking" benefit from enabling CoT (e.g., Gemini-2.5-Flash-Thinking). When a model already exhibits strong inherent reasoning, additional explicit CoT can reduce performance, as seen in o4-mini at medium/high reasoning depths. Open-source models do not consistently benefit from CoT without targeted training for reasoning behaviors, suggesting a direction for future supervised and RL post-training.

| Model | CoT (Accuracy %) | Non-CoT (Accuracy %) |
|---|---|---|
| Gemini-2.5-Pro | 50.67% | 52.33% |
| Gemini-2.5-Flash-no-think | 42.00% | 40.00% |
| Gemini-2.5-Flash-Thinking | 52.00% | 46.00% |
| o4-mini (low) | 52.00% | 51.00% |
| o4-mini (medium) | 50.67% | 53.00% |
| o4-mini (high) | 50.33% | 54.33% |

Table 16: Effect of explicit chain-of-thought (CoT) prompting on MSEarth-Bench-mini. Higher is better; values are accuracy (%). For o4-mini, low/medium/high denote shorter-to-longer reasoning traces.

| Model | Atmospheric Sciences | | | Ecology and Biosciences | | |
|---|---|---|---|---|---|---|
| | Meteor. | Climat. | Atmos. RS | Ecosys. Ecol. | Landsc. Ecol. | Aquat. Ecol. |
| InternVL-8B | 0.4039 | 0.3643 | 0.5238 | 0.3833 | 0.4792 | 0.3030 |
| InternVL-78B | 0.4624 | 0.3643 | 0.5476 | 0.6333 | 0.5208 | 0.6061 |
| InternVL3-78B | 0.4847 | 0.3857 | 0.4762 | 0.6333 | 0.5417 | 0.6364 |
| Qwen2.5-VL-72B | 0.4903 | 0.4071 | 0.4524 | 0.4500 | 0.4375 | 0.5455 |
| Claude-3.7-Sonnet | 0.5599 | 0.4714 | 0.6429 | 0.6500 | 0.6042 | 0.5758 |
| Gemini-2.5-Pro-Thinking | 0.5877 | 0.5071 | 0.5714 | 0.7000 | 0.5833 | 0.6667 |
| GPT-4o | 0.5097 | 0.4429 | 0.5714 | 0.5667 | 0.5208 | 0.6667 |
| GPT-4o-mini | 0.4457 | 0.3857 | 0.5000 | 0.6333 | 0.4375 | 0.5455 |
| Gemini-2.5-Flash-Thinking | 0.5265 | 0.3929 | 0.5476 | 0.7500 | 0.5833 | 0.6364 |
| Gemini-2.5-Flash | 0.5153 | 0.4000 | 0.5952 | 0.6167 | 0.5625 | 0.5455 |
| Intern-S1 | 0.6320 | 0.5870 | 0.6750 | 0.7350 | 0.6580 | 0.7020 |
| Intern-S1-mini-MSEarth | 0.6080 | 0.5630 | 0.6420 | 0.7120 | 0.6350 | 0.6780 |
| Intern-S1-mini | 0.5850 | 0.5310 | 0.6180 | 0.6870 | 0.6090 | 0.6450 |

Table 17: Model Performance on Primary and Sub-Disciplines of Earth Science (Accuracy)

## L    MORE RESULTS

For MCQ and OE questions, we used radar charts to illustrate the performance of various models across different disciplines. We also give some case studies in Figure 17 and Figure 18. We also present detailed performance breakdowns of all models across every sub-discipline of Earth science in Tables 20.

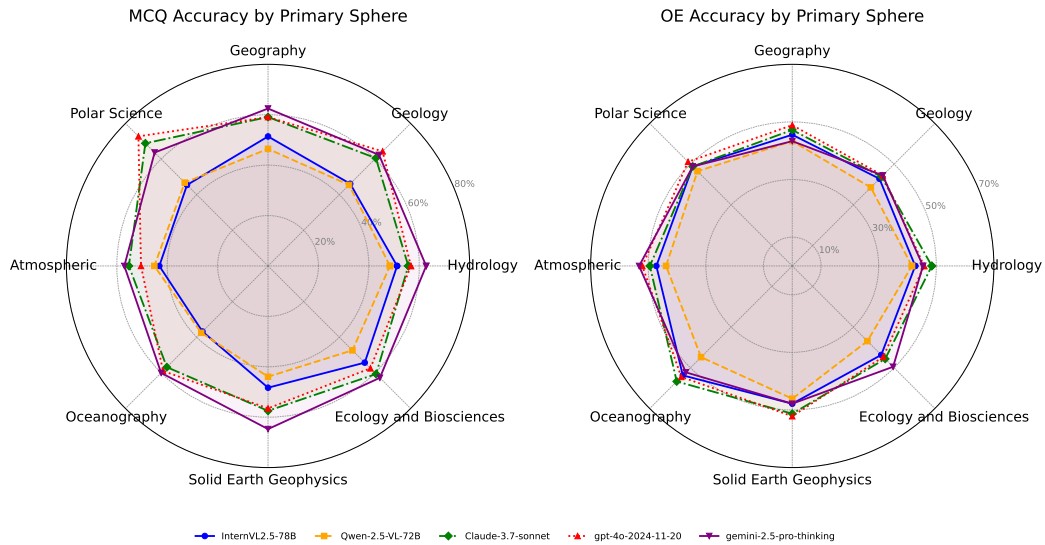

Figure 16: Performance comparison of different models across various subjects.

| Model | Geography | | | Geology | | |
|---|---|---|---|---|---|---|
| | Phys. Geog. | Urban Geog. | Reg. Geog. | Sediment. | Struct. Geol. | Quat. Geol. |
| InternVL-8B | 0.4765 | 0.4444 | 0.4444 | 0.5090 | 0.4268 | 0.4878 |
| InternVL-78B | 0.5451 | 0.4815 | 0.5556 | 0.5663 | 0.4634 | 0.5610 |
| InternVL3-78B | 0.5884 | 0.3333 | 0.6667 | 0.5986 | 0.5366 | 0.6585 |
| Qwen2.5-VL-72B | 0.5307 | 0.3704 | 0.6667 | 0.5305 | 0.5000 | 0.6098 |
| Claude-3.7-Sonnet | 0.5993 | 0.4074 | 0.8889 | 0.6057 | 0.6341 | 0.4878 |
| Gemini-2.5-Pro-Thinking | 0.6354 | 0.5556 | 0.5556 | 0.6487 | 0.5854 | 0.6341 |
| GPT-4o | 0.5884 | 0.5556 | 0.7778 | 0.6703 | 0.6341 | 0.6829 |
| GPT-4o-mini | 0.5090 | 0.4815 | 0.6667 | 0.5269 | 0.5488 | 0.5122 |
| Gemini-2.5-Flash-Thinking | 0.6426 | 0.5556 | 0.7778 | 0.5986 | 0.5976 | 0.5122 |
| Gemini-2.5-Flash | 0.5921 | 0.5185 | 0.7778 | 0.5771 | 0.5366 | 0.5854 |
| Intern-S1 | 0.6830 | 0.6250 | 0.9120 | 0.6970 | 0.6780 | 0.7250 |
| Intern-S1-mini-MSEarth | 0.6590 | 0.5980 | 0.8850 | 0.6730 | 0.6540 | 0.6980 |
| Intern-S1-mini | 0.6320 | 0.5640 | 0.8530 | 0.6450 | 0.6210 | 0.6670 |

Table 18: Model Performance on Primary and Sub-Disciplines of Earth Science (Accuracy, Continued)

| Model | Hydrology | | | Oceanography | | |
|---|---|---|---|---|---|---|
| | River Hydrol. | Groundw. Hydrol. | Limnol. | Ocean Phys. | Ocean Geol. | Env. Oceanogr. |
| InternVL-8B | 0.4550 | 0.4297 | 0.4348 | 0.3800 | 0.4762 | 0.2941 |
| InternVL-78B | 0.5500 | 0.4688 | 0.5652 | 0.4800 | 0.5714 | 0.3529 |
| InternVL3-78B | 0.5400 | 0.4766 | 0.5652 | 0.5250 | 0.6190 | 0.5882 |
| Qwen2.5-VL-72B | 0.5850 | 0.5078 | 0.5435 | 0.4900 | 0.6667 | 0.3529 |
| Claude-3.7-Sonnet | 0.6150 | 0.5391 | 0.5435 | 0.5650 | 0.7143 | 0.3529 |
| Gemini-2.5-Pro-Thinking | 0.6700 | 0.6172 | 0.6739 | 0.5750 | 0.6667 | 0.5882 |
| GPT-4o | 0.6200 | 0.5625 | 0.5217 | 0.5800 | 0.7619 | 0.4706 |
| GPT-4o-mini | 0.5900 | 0.4766 | 0.5435 | 0.4650 | 0.5238 | 0.4706 |
| Gemini-2.5-Flash-Thinking | 0.6250 | 0.5625 | 0.5435 | 0.5000 | 0.6667 | 0.4706 |
| Gemini-2.5-Flash | 0.5700 | 0.4922 | 0.6304 | 0.4800 | 0.5238 | 0.3529 |
| Intern-S1 | 0.7050 | 0.6680 | 0.6970 | 0.6320 | 0.7950 | 0.6230 |
| Intern-S1-mini-MSEarth | 0.6820 | 0.6430 | 0.6720 | 0.6080 | 0.7680 | 0.5950 |
| Intern-S1-mini | 0.6570 | 0.6150 | 0.6450 | 0.5830 | 0.7320 | 0.5680 |

Table 19: Model Performance on Primary and Sub-Disciplines of Earth Science (Accuracy, Continued)

| Model | Polar Science | | | Solid Earth Geophysics | | |
|---|---|---|---|---|---|---|
| | Glaciol. | Permafrost Sci. | Polar Ocean | Seismol. | Tectonophys. | Geomagn. |
| InternVL2.5-8B | 0.4571 | 0.7500 | 0.0000 | 0.4248 | 0.5625 | 0.5455 |
| InternVL2.5-78B | 0.4571 | 0.5000 | 1.0000 | 0.4902 | 0.4375 | 0.5455 |
| InternVL3-78B | 0.5286 | 0.5000 | 1.0000 | 0.5033 | 0.5000 | 0.6364 |
| Qwen2.5-VL-72B | 0.6286 | 0.5000 | 1.0000 | 0.4706 | 0.3750 | 0.7273 |
| Claude-3.7-Sonnet | 0.7000 | 0.5000 | 1.0000 | 0.5752 | 0.5625 | 0.6364 |
| Gemini-2.5-Pro-Thinking | 0.6286 | 1.0000 | 1.0000 | 0.6863 | 0.5000 | 0.8182 |
| GPT-4o | 0.7286 | 0.7500 | 1.0000 | 0.5163 | 0.6250 | 0.9091 |
| GPT-4o-mini | 0.5571 | 0.7500 | 1.0000 | 0.4379 | 0.4375 | 0.6364 |
| Gemini-2.5-Flash-Thinking | 0.6571 | 0.7500 | 1.0000 | 0.5359 | 0.6250 | 0.8182 |
| Gemini-2.5-Flash | 0.6812 | 0.7500 | 1.0000 | 0.6013 | 0.5000 | 0.5455 |
| Intern-S1 | 0.7650 | 1.0000 | 1.0000 | 0.7230 | 0.6850 | 0.9320 |
| Intern-S1-mini-MSEarth | 0.7380 | 0.9500 | 1.0000 | 0.6970 | 0.6580 | 0.9050 |
| Intern-S1-mini | 0.7120 | 0.9000 | 1.0000 | 0.6650 | 0.6230 | 0.8780 |

*Note:* 1. Sub-disciplines listed are the top 3 with the largest sample size in each primary discipline;
2. Abbreviations: Meteor.=Meteorology, Climat.=Climatology, Atmos. RS=Atmospheric Remote Sensing,
Ecosys. Ecol.=Ecosystem Ecology, Landsc. Ecol.=Landscape Ecology, Aquat. Ecol.=Aquatic & Limnological Ecology,
Phys. Geog.=Physical Geography, Reg. Geog.=Regional Geography, Sediment.=Sedimentology,
Struct. Geol.=Structural Geology, Quat. Geol.=Quaternary Geology, River Hydrol.=River & Estuarine Hydrology,
Groundw. Hydrol.=Groundwater Hydrology, Limnol.=Limnology, Env. Oceanogr.=Environmental Oceanography,
Glaciol.=Glaciology, Seismol.=Seismology, Tectonophys.=Tectonophysics, Geomagn.=Geomagnetism.

Table 20: Model Performance on Primary and Sub-Disciplines of Earth Science (Accuracy, Continued)

---

**Multiple Choice VQA Case Study**

**Original Caption**: (a) Temporal distribution of precipitation, wind direction, and wind intensity (maximum and mean) on 27 February observed at the weather station at Navarredonda (see Fig. 1b); (b) spatial pattern of precipitation (isolines in mm) observed on 27 February over the study area (circle, Navarredonda automatic weather station).

**Question:** What does the temporal distribution of precipitation in Figure 9(a) indicate about the morning hours of February 27?

**Options:**
A. Minimal precipitation occurred.
B. A brief period of heavy precipitation occurred.
C. Precipitation was evenly distributed throughout the morning.
D. Precipitation was absent in the morning.

**Response:** B. A brief period of heavy precipitation occurred.

**Refined caption**: (a) Temporal distribution of precipitation, wind direction, and wind intensity (maximum and mean) on 27 February as observed at the Navarredonda weather station. The data highlights a brief period of increased instability indices, Froude number, and meridional moisture flux during the morning and central hours, attributed to the short-lived flow perpendicular to the mountain range. (b) Spatial pattern of precipitation on 27 February over the study area, with isolines indicating precipitation levels in millimeters. The pattern confirms a broad area of precipitation with measured values between 30 mm and higher, with maximum values observed around the Navarredonda automatic weather station.

**Reasoning chain:**
Step 1: Observe the precipitation graph in Figure (a), which shows bar heights representing precipitation recorded in 10-minute intervals.
Step 2: Analyze the morning hours (e.g., 06Z to 09Z); the bar graph shows a significant increase in precipitation during this time.
Step 3: Conclude that there was a brief period of heavy precipitation in the morning.

---

`qwen-2.5-VL-instruct` ❌

Answer: A
Explanation: The top graph in Figure (a) shows the temporal distribution of precipitation in 10-minute intervals. In the morning hours (from 00:00 UTC to around 08:00 UTC), the bars representing precipitation are relatively low, indicating minimal precipitation during this time frame. There is no evidence of heavy precipitation or even distribution throughout the morning, nor is there an indication that precipitation was completely absent.

`gemini-2.5-pro-thinking` ✅

Answer: B
Explanation: Figure (a) shows the temporal distribution of precipitation. The x-axis represents time in UTC, and the y-axis represents precipitation in mm/10min. The morning hours of February 27 can be considered as the period from approximately 06:00 UTC to 12:00 UTC. Observing the precipitation graph (top panel) during these hours:\n1. Precipitation is clearly present throughout this period, so option D ('Precipitation was absent in the morning') is incorrect.\n2. The amount of precipitation is not minimal; there are consistent bars, many of which are around 0.5 mm/10min to 1 mm/10min, and some even higher. Thus, option A ('Minimal precipitation occurred') is incorrect.\n3. The height of the precipitation bars varies significantly throughout the morning, indicating that the precipitation was not evenly distributed. For instance, there are peaks and troughs in intensity. Therefore, option C ('Precipitation was evenly distributed throughout the morning') is incorrect.\n4. Between 06:00 UTC and 07:00 UTC, there is a noticeable increase in precipitation intensity, with several 10-minute intervals showing precipitation rates of 1 mm/10min or more, and one peak reaching approximately 1.5 mm/10min around 06:30 UTC. This can be characterized as a period of heavier precipitation compared to other parts of the morning (e.g., 08:00-10:00 UTC where precipitation is generally lower). This period of higher intensity is relatively brief in the context of the entire morning. Another similar period of increased intensity occurs around 11:00-12:00 UTC. Therefore, option B ('A brief period of heavy precipitation occurred') accurately describes the observed pattern.

`claude-3-7-sonnet-thinking` ❌

Answer: C. Precipitation was evenly distributed throughout the morning.
Explanation: Looking at Figure (a), the top panel shows precipitation data for February 27, 2010, with the x-axis representing hours from 00 to 24 UTC. During the morning hours (approximately 06-12 UTC), the blue precipitation bars display a relatively consistent pattern with similar heights. There are no major gaps in precipitation or extremely intense brief episodes during this period. Instead, the precipitation appears to be occurring steadily throughout the morning hours with a fairly even distribution, making option C the correct answer.

`gpt-4o-2024-11-20` ✅

Answer: B. A brief period of heavy precipitation occurred.
Explanation: Figure (a) shows the temporal distribution of precipitation with vertical bars indicating precipitation levels over 10-minute intervals. During the morning hours of February 27 (especially between 06:00 UTC and 08:00 UTC), there is a noticeable spike in precipitation levels, indicating a brief period of heavy precipitation compared to other time intervals.

Figure 17: Case Study of Multiple Choice VQA.

**Open-Ended VQA Case Study 1**

**Original Caption**: Schematic illustration of locations of the photometric stations.

**Question**: Which station lies in a semi-arid steppe zone?

**Response**: Baruun-Urt

**Reasoning chain:**

Step 1: From the Figure, Baruun-Urt is outside the Gobi Desert's shaded area, suggesting it is not in the arid region.

Step 2: The semi-arid steppe zone is consistent with areas having denser vegetation, implied by Baruun-Urt's location in the Figure.

**Refined caption**: Schematic illustration of the locations of the photometric stations used for atmospheric AOD measurements in the Eastern Gobi region of Mongolia. The stations are situated at approximately the same latitude and include Sainshand (located in an arid zone), Dalanzadgad (also in an arid zone), and Baruun-Urt (located in a semi-arid steppe zone with denser vegetation cover).

`qwen-2.5-VL-instruct` ❌

Answer: Dalanzadgad
Explanation: Located in semi-arid area

`gemini-2.5-pro-thinking` ✅

Answer: Baruun-Urt
Explanation: The Gobi Desert is an arid zone. Baruun-Urt is located north of the Gobi Desert, in an area likely to be a semi-arid steppe transition zone.

`claude-3-7-sonnet` ✅

Answer: Baruun-Urt
Explanation: Located at northern edge of Gobi Desert region where desert transitions to semi-arid steppe ecological zone.

`gpt-4o-2024-11-20` ❌

Answer: Sainshand,
Explanation: The map shows Sainshand located within the semi-arid steppe zone of the Gobi Desert.

**Open-Ended VQA Case Study 2**

**Original Caption**: Map of Gösing with marked earth buildings from the cadastre.

**Question**: Which natural feature primarily impacts settlement patterns visible in the map?

**Response**: Terrain slope

**Reasoning chain:**

Step 1: Observing the map shows that areas near slopes have higher earth building clusters.

Step 2: Caption reveals that terrain influences settlement and construction choices.

Step 3: Terrain slope emerges as the most significant natural factor influencing patterns.

**Refined caption:** Detailed map of Gösing illustrating the distribution of earth buildings as recorded in the earth building cadastre. The map highlights how the topography of the terrain influences the settlement patterns of these structures, providing insight into the construction techniques and methods that shape the appearance of entire streets within the village.

`gemini-2.5-pro-thinking` ✅

Answer: Terrain or topography
Eexplanation: The map shows buildings (settlements) concentrated along roads, and the presence of dashed lines, likely representing contour lines, suggests varying elevation. Settlements often follow flatter areas or valleys in hilly terrain for easier construction and access. Therefore, the terrain/topography is the primary natural feature influencing where settlements are located.

`qwen-2.5-VL-instruct` ❌

Answer: River proximity
Explanation: Settlements cluster near river.

`gpt-4o-2024-11-20` ✅

Answer: Terrain elevation differences
Explanation: The map shows settlement patterns concentrated along roads and flatter areas, avoiding steeper or elevated terrain, which is typical in areas with significant elevation differences.

Figure 18: Case Study of Open-Ended VQA.

