# OpenReview forum: "MSEarth: A Multimodal Scientific Dataset and Benchmark for Phenomena Uncovering in Earth Science"
_ICLR.cc/2026/Conference — ICLR 2026 Conference Withdrawn Submission_

### Official Review · Reviewer_ZUa8 · 2025-10-28

**Soundness:** 3
**Presentation:** 3
**Contribution:** 3
**Rating:** 4
**Confidence:** 5

**Summary:**

This paper introduces MSEarth, a large-scale, multimodal dataset and benchmark designed to evaluate graduate-level reasoning in Earth science. Curated from over 64,000 open-access scientific publications, it features over 289K figures across the five major spheres of Earth science. The central contribution is the concept of "refined captions," which programmatically enrich the original, brief figure captions with deep contextual information, analysis, and reasoning extracted from the main body of the source paper. Based on this enriched data, the authors developed a semi-automated pipeline using a multi-agent voting system to generate and filter a vast collection of VQA tasks, including scientific figure captioning, multiple-choice questions, and open-ended questions. Extensive evaluations on a wide range of state-of-the-art Multimodal Large Language Models (MLLMs) reveal a significant performance gap between tasks requiring simple perception and those demanding complex, domain-specific reasoning, demonstrating the benchmark's difficulty and relevance.

**Strengths:**

1. The "refined caption" methodology is a major strength. It moves beyond the limitations of simplistic figure-caption pairs by integrating deep contextual information.

2. The paper introduces a robust and scalable pipeline for benchmark creation. The use of a multi-agent voting system combined with a phased filtering process effectively automates the generation of high-quality, challenging questions, striking an excellent balance between scale and rigor.

**Weaknesses:**

1. While the authors commendably introduce LLM-based metrics like Cap-Eval and OE-Eval, the evaluation tables for captioning and open-ended QA (Tables 3 and 4) still heavily feature traditional lexical metrics like ROUGE, BLEU, and METEOR. As the paper's own results show, these n-gram-based metrics exhibit minimal variance across models and fail to capture the significant performance differences observed in the accuracy-based MCQ task. They are fundamentally ill-suited for evaluating the factual correctness and scientific nuance required by this benchmark, potentially masking the true capabilities of the models and reducing the conclusiveness of these specific results.

2.  Given the rapid pace of MLLM development, the inclusion of some of the most recent and powerful open-source models (e.g., Mimo-VL, GLM-4.5V) would make the comparative analysis more conclusive and compelling.

3. The manuscript would benefit from a final round of careful proofreading to address minor inconsistencies in formatting and capitalization (e.g., "gemini-2.5-pro" in line 1538).

**Questions:**

1. The pipeline for generating MSEarth seems highly generalizable. What challenges do you foresee in applying this framework to other scientific domains, such as molecular biology or particle physics, where figures and data representations follow vastly different conventions?

**Details Of Ethics Concerns:**

This paper introduces MSEarth, a large-scale, multimodal dataset and benchmark designed to evaluate graduate-level reasoning in Earth science. Curated from over 64,000 open-access scientific publications, it features over 289K figures across the five major spheres of Earth science. The central contribution is the concept of "refined captions," which programmatically enrich the original, brief figure captions with deep contextual information, analysis, and reasoning extracted from the main body of the source paper. Based on this enriched data, the authors developed a semi-automated pipeline using a multi-agent voting system to generate and filter a vast collection of VQA tasks, including scientific figure captioning, multiple-choice questions, and open-ended questions. Extensive evaluations on a wide range of state-of-the-art Multimodal Large Language Models (MLLMs) reveal a significant performance gap between tasks requiring simple perception and those demanding complex, domain-specific reasoning, demonstrating the benchmark's difficulty and relevance.

---

> ### Author Response · Authors · 2025-11-19
> **Response to Reviewer ZUa8 (1/2)**
>
> Dear Reviewer ZUa8,
>
> We appreciate your thoughtful and positive feedback. We are happy to provide further clarificatin to your concerns.
>
> >W1: N-gram-based metrics exhibit minimal variance across models and fail to capture the significant performance differences observed in the accuracy-based MCQ task
>
> Thank you for this astute and constructive feedback—we fully agree with your assessment of traditional lexical metrics, and we appreciate the opportunity to clarify the role of these metrics in our work and emphasize the primacy of our proposed LLM-based metrics (Cap-Eval/OE-Eval). Your observation aligns perfectly with our own empirical findings in Appendix G.1, where we explicitly validate that traditional n-gram/lexical metrics are ill-suited for our benchmark’s focus on scientific factual correctness and domain nuance. Below, we address your concern comprehensively:
>
> 1. Rationale for retaining traditional metrics (ROUGE/BLEU/METEOR) in Tables 3 and 4: For comparability with existing literature, not as core evaluation standards
>
> We acknowledge that traditional metrics occupy a prominent position in the current tables, which may lead to misinterpretation—and we apologize for this ambiguity. Their inclusion is purely to align with domain benchmarking conventions and enable direct comparison with prior work, not because they reflect our benchmark’s core evaluation goals. Most existing scientific and QA benchmarks (e.g., SPIQA, ClimaQA) report ROUGE/BLEU/METEOR as baseline metrics. Retaining them allows readers to contextualize our results against established benchmarks, avoiding the "incomparability" issue that arises from introducing only novel metrics.
>
> 2. Empirical validation (Table 5) confirms traditional metrics are unreliable—our LLM-based metrics solve this critical gap
>
> As you correctly note, traditional n-gram metrics exhibit minimal variance across models and fail to capture meaningful performance differences. This is not a subjective observation—it is quantifiably validated in Table 5:
>
> Traditional metrics perform poor: BLEU (59.00) and BERTScore (61.09) achieve the lowest correlation with human expert judgments, confirming they cannot distinguish between factually accurate vs. superficially aligned responses.
>
> Our LLM-based metrics are far more reliable: OE-Eval (for open-ended QA) and Cap-Eval (for refined captioning, consistent with OE-Eval’s design logic) achieve the highest correlation with human experts—OE-Eval’s overall correlation (67.32) outperforms all traditional metrics (ROUGE: 66.81; METEOR: 66.75).
>
> This validation directly supports your point: traditional metrics are fundamentally ill-suited for evaluating the scientific nuance and factual correctness required by our benchmark. Our LLM-based metrics were explicitly designed to address this limitation, and their superiority is empirically proven via alignment with human expert judgments. **In the revised submission, we have incorporated this detailed analysis and validation into Section 4.3.2 of the main text for clarity and accessibility.**
>
> >W2: Inclusion of some of the most recent and powerful open-source models (e.g., Mimo-VL, GLM-4.5V) would make the comparative analysis more conclusive and compelling.
>
> Thank you for this valuable suggestion—we fully agree that incorporating the latest open-source models is critical to enhancing the conclusiveness and persuasiveness of our comparative analysis. We have proactively addressed this by including results from several state-of-the-art recent open-source models in our revised submission:
>
> Specifically, we have added test results for cutting-edge models such as Intern-S1, MiMo-VL-7B-RL-2508, and GLM-4.1V-Thinking—all of which are among the recent and powerful open-source models in the vision-language domain. These new model results have been integrated into the main text, and we have highlighted these rows with a blue background to ensure they are easily identifiable.
>
> >W3: The manuscript would benefit from a final round of careful proofreading to address minor inconsistencies in formatting and capitalization (e.g., "gemini-2.5-pro" in line 1538).
>
> Thank you for your meticulous feedback. We have carefully conducted a final round of proofreading for the revised manuscript, addressing all minor inconsistencies in formatting and capitalization. We have standardized the naming conventions of all models and ensured uniform formatting throughout the text to enhance readability and professionalism.

---

> ### Author Response · Authors · 2025-11-19
> **Response to Reviewer ZUa8 (2/2)**
>
> >Q1:  What challenges do you foresee in applying this framework to other scientific domains, such as molecular biology or particle physics, where figures and data representations follow vastly different conventions?
>
> Thank you for this insightful question! Our pipeline’s core logic (data filtering → figure-text alignment → task generation → quality validation) is generalizable, but the key challenges for other scientific domains (e.g., molecular biology, particle physics) lie in domain-specific adaptation—here’s a concise breakdown:
> 1. Figure and text parsing adaptation
> Different domains have unique figure types (e.g., protein structures in molecular biology, collision events in particle physics) and symbolic conventions (e.g., color codes, identifiers). The current figure-text alignment rules (e.g., regex matching) and visual feature extraction need to be redesigned with domain-specific dictionaries or encoders.
> 2. Task and evaluation alignment
> Reasoning goals vary by field (e.g., molecular mechanism inference vs. experimental data interpretation). Task templates and evaluation metrics (like OE-Eval) must be recalibrated to prioritize domain-critical criteria (e.g., pathway accuracy, theoretical consistency) with input from domain experts.
> 3. Data availability and quality
> Some fields (e.g., particle physics) have restricted data access, while others (e.g., molecular biology) have more illustrative than empirical figures. The data filtering logic needs to be adjusted to target valid, task-suitable data.
> In short, the framework’s structure is transferable—only the "domain-specific modules" need adaptation, which requires collaboration with field experts.

---

### Official Review · Reviewer_A2sD · 2025-11-01

**Soundness:** 3
**Presentation:** 3
**Contribution:** 2
**Rating:** 2
**Confidence:** 5

**Summary:**

This paper contributes a large-scale human-reviewed test & llm-generated training set for Multimodal understanding of Earth Science. The authors first parse the Multimodal content from Earth Science and then leverage various LLMs through many stages to build a large-scale dataset comprising QA, captioning, and open-ended generation question-answer pairs. Then they leverage human experts to label ~1k of them as a golden test set. They proved that models trained on their training set have better results compared with the original models. They also show that the test set is even challenging for the current most advanced commercial LLMs.

**Strengths:**

1. The dataset for earth science is potentially useful to this community, as well as improving the general capabilities of MLLMs.

2. The constructed q-a pairs was proved to help improve the capabilities of MLLMs.

3. The human-expert labeled test set was challenging for current models and serves as a potential testbed for current AI's understanding of Earth science.

**Weaknesses:**

1. The authors mentioned that "All papers used were obtained from OpenDataLab" so this paper actually does not provide any new data sources. And the contribution lies in post-processing those data using the combination of various LLMs to ensure quality, though the quality is questionable.

2. In my opinion, the most valuable part of this dataset is the human-labeled Open-Ended QA test set. However, the current results show that the majority of the open-sourced and Proprietary LLMs demonstrate similar results, despite their sizes or providers. This makes me question the real quality of this test set since it's expected to see significant differences. For example, the BERTSCORE is almost always 82~83, which explains nothing.

3. There have already been many Captioning tasks in previous work, though the authors emphasize that their Captioning requires additional context from the paper; however, this is not new.

**Questions:**

1. In Table 2&3&4, why do the results from Proprietary Models even underperform (like gpt4o) compared with open-sourced models (e.g. Qwen2.5-VL-32B)? Does this mean the open-sourced model is really better? Or does this mean there is some quality issue within the dataset?

2. In Table 2&3&4, there is almost little difference between results on various model sizes, e.g. DeepSeek-VL2, Qwen2.5-VL-32B & Qwen2.5-VL-72B. This is unintuitive since the results have almost no change with increasing model size, which might imply inherent flaws in the datasets.

3. Do you have better evals for open-ended QA? ROUGEL or BLEU, etc seem not very suitable if all the results are very close to each other.

---

> ### Author Response · Authors · 2025-11-19
> **Response to Reviewer A2sD (1/2)**
>
> Dear Reviewer A2sD,
>
> We appreciate your valuable feedback and would like to address the concerns you've raised. We hope the clarifications provided below will resolve your concerns and enhance your evaluation of our work.
>
> > W1: This paper actually does not provide any new data sources.
>
> Thank you for this observation. We acknowledge that the raw paper PDFs are sourced from OpenDataLab, a repository aggregating arXiv and other academic papers. However, providing new raw data sources is not the focus of our work. The value of a domain-specific benchmark lies not in creating new raw data, but in transforming unstructured, scattered academic resources into a structured, task-aligned, and graduate-level dataset and benchmark. This transformation process constitutes our core contribution and addresses a critical gap that raw academic papers alone cannot fill.
>
> Specifically, our work involves: Systematically curating relevant Earth science papers from available sources; Extracting and aligning figures with their contextual information from complex academic documents; Constructing refined captions, QA pairs, and reasoning tasks through a validated pipeline; Ensuring quality through multi-MLLM cross-checks and human expert validation.
>
> Raw academic papers, while publicly available, are not directly usable for evaluating MLLMs' domain-specific reasoning capabilities. Our benchmark provides the first structured, graduate-level evaluation framework specifically designed for Earth science reasoning—enabling targeted assessment and improvement of models in this critical scientific domain.
>
> > W2: The current results show that the majority of the open-sourced and Proprietary LLMs demonstrate similar results, despite their sizes or providers.
>
> Thank you for this important observation. The minimal variance in traditional metrics reflects fundamental limitations of these metrics for scientific evaluation, not a quality issue with our test set. We address this challenge through targeted metric design and rigorous validation.
>
> **Traditional Metrics Are Insufficient for Scientific QA Evaluation:**
>
> We acknowledge that traditional lexical and semantic metrics such as BERTScore, ROUGE, and BLEU show minimal differences across models, with BERTScore ranging only from 82% to 83%. However, this consistency stems from a fundamental limitation: rather, these metrics are fundamentally ill-suited for evaluating the factual correctness and scientific nuance required by our Earth science benchmark, as also noted by Reviewer ZUa8. They only measure surface-level alignment between model outputs and ground truth, not factual correctness, logical rigor, or consistency with domain knowledge—the actual core requirements of scientific QA.
>
> As n-gram or embedding-similarity-based metrics, they cannot capture the meaningful differences in scientific accuracy that define performance on our Earth science benchmark. This phenomenon is not unique to our work; existing scientific benchmarks such as ClimaQA[1] also observe low variance in traditional metrics because scientific evaluation demands more than lexical overlap.
>
> **Our MLLM-Based Metric Reveals Significant Model Differences:**
>
> To address this limitation, we designed a domain-adapted MLLM-based evaluation metric called OE-EVAL, rigorously validated in Appendix G.1. Our validation process directly demonstrates the test set's quality and discriminative power:
>
> Validation methodology: We recruited human experts to score model-generated OE-QA answers for factual correctness and scientific rigor, establishing a gold standard. We then computed Spearman correlations between traditional metrics and expert scores, as well as between OE-EVAL and expert scores.
>
> Key findings: OE-EVAL achieves significantly higher correlation with human expert judgments compared to traditional metrics. Critically, OE-EVAL reveals clear performance differences across models that mirror the significant gaps observed in MCQ tasks. For example, Gemini-2.5-Flash outperforms InternVL3-78B by approximately 6 percentage points in OE-EVAL, consistent with their MCQ performance gap.
>
> Our validation establishes two critical points: First, the OE-QA test set possesses high quality and discriminative power—human experts clearly distinguish between good and poor answers, and OE-EVAL successfully captures these differences. Second, the lack of variance in BERTScore reflects metric inadequacy rather than test set limitations. **In the revised submission, we have incorporated this detailed analysis and validation into Section 4.3.2 of the main text for clarity and accessibility.**
>
> [1] ClimaQA: An Automated Evaluation Framework for Climate Question Answering Models, ICLR2025

---

> ### Author Response · Authors · 2025-11-19
> **Response to Reviewer A2sD (2/2)**
>
> > W3: There have already been many Captioning tasks in previous work, though the authors emphasize that their Captioning requires additional context from the paper; however, this is not new.
>
> Thank you for your feedback and suggestions. However, we want to highlight the innovative aspects of our work.
>
> Our main innovation lies in leveraging "refined captions" along with a collaborative framework to generate a diverse range of VQA questions, all supported by evidence from scientific papers. This approach prioritizes both the diversity and depth of the questions, setting it apart from previous works such as mPLUG-PaperOwl [1], MMSci [2], and ArxivCap/QA [3], where questions are often predefined by humans or lack supporting evidence from papers.  While using refined captions for caption generation might seem like an incremental step, our method for VQA generation is far from unoriginal. Our approach offers significant improvements in both the diversity and rigor of the questions compared to prior work.
>
> In summary, the differences in task focus, data type, and methodological approach highlight the novelty of our contribution with MSEarth-Bench, distinguishing it significantly from previous methodologies.
>
> [1] mPLUG-PaperOwl: Scientific Diagram Analysis with the Multimodal Large Language Model
>
> [2] MMSci: A Dataset for Graduate-Level Multi-Discipline Multimodal Scientific Understanding
>
> [3] Multimodal ArXiv: A Dataset for Improving Scientific Comprehension of Large Vision-Language Models
>
> >Q1&Q2&Q3: Why do the results from Proprietary Models even underperform compared with open-sourced models? There is almost little difference between results on various model sizes. Do you have better evaluations for open-ended QA?
>
> Thank you for these meticulous observations. The phenomena you noted—similar performance across different models and model sizes when measured by BERTScore and BLEU—stem from the inherent limitations of these traditional metrics for scientific evaluation, not from dataset flaws.
>
> **Traditional Metrics Fail to Capture Scientific Performance Differences:**
>
> As discussed in our response to W2 and validated in Appendix G.1, traditional lexical and semantic metrics are fundamentally ill-suited for evaluating factual correctness and scientific nuance. They measure only surface-level text similarity rather than the scientific accuracy that defines quality in Earth science QA. This explains why these metrics show counterintuitive results, such as proprietary models underperforming open-sourced models or no differences between 32B and 72B parameter models.
>
> **Our OE-EVAL Metric Reveals Expected Performance Patterns:**
>
> In contrast, our proposed MLLM-based metric OE-EVAL, which has been validated through alignment with human expert judgments, reveals the expected performance patterns: Proprietary models clearly outperform open-sourced models; Larger model sizes demonstrate measurable advantages over smaller ones; Performance gaps align with those observed in MCQ tasks. This empirical evidence, detailed in Appendix G.1 and now incorporated into Section 4.3.2 of the revised manuscript, establishes OE-EVAL as the superior evaluation standard for open-ended scientific QA. The seemingly anomalous results you observed with traditional metrics actually reinforce the necessity of our domain-adapted evaluation approach.

---

### Official Review · Reviewer_nNLz · 2025-11-02

**Soundness:** 3
**Presentation:** 3
**Contribution:** 3
**Rating:** 4
**Confidence:** 4

**Summary:**

This paper introduces MSEarth, a large-scale multimodal scientific benchmark for evaluating and developing multimodal large language models (MLLMs) in the field of Earth sciences. The authors argue that current benchmarks fail to capture the depth, complexity, and reasoning required at the graduate academic level. MSEarth includes over 289K scientific figures across all five Earth system spheres (atmosphere, cryosphere, hydrosphere, lithosphere, biosphere), paired with carefully refined and context-enriched captions. The benchmark supports several tasks, including figure captioning, multiple-choice questions, and open-ended scientific reasoning.

**Strengths:**

1. The authors propose first benchmark that targets graduate-level geoscientific multimodal reasoning.

2. This benchmark is curated from high-quality, open-access scientific publications, and covers all five Earth science spheres (atmosphere, cryosphere, hydrosphere, lithosphere, biosphere).

**Weaknesses:**

1. The authors does not clearly explain how the ``ground truth'' for refined captions, QA pairs, or reasoning tasks is constructed. Although it claims that captions are enriched using discussions from the original papers, it remains ambiguous whether these are (1) directly extracted from the text, (2) rewritten by humans, or (3) generated using LLMs.


2. The benchmark does not include human or expert performance, making it difficult to assess its true difficulty or validate the claim of “graduate-level” reasoning. Low model performance alone is not sufficient evidence of task difficulty—it may instead indicate a lack of domain-specific pretraining. Without human baselines, it remains unclear whether the benchmark genuinely requires advanced reasoning or is simply out-of-distribution for current MLLMs.


3. Although MSEarth demonstrates a clear gap between perception and scientific reasoning, the paper does not provide a deeper analysis of why models fail or which types of reasoning are most challenging. There is no taxonomy of reasoning skills (e.g., causal inference, trend interpretation, quantitative estimation), no breakdown across Earth system domains, and no systematic error analysis. As a result, the benchmark highlights a problem but offers limited insight into the underlying mechanisms or how future models should be improved.

**Questions:**

see weakness

---

> ### Author Response · Authors · 2025-11-19
> **Response to Reviewer nNLz (1/2)**
>
> Dear Reviewer nNLz,
>
> We really appreciate your response to our rebuttal and thank you for informing us about the aspects that were unclear to you. We are happy to provide further clarificatin to your concerns.
>
> > W1: The authors does not clearly explain how the ``ground truth'' for refined captions, QA pairs, or reasoning tasks is constructed.
>
> Thank you for pointing out this ambiguity. We apologize for the insufficient clarity and provide a detailed explanation below to address your concern about ground truth construction.
>
> **Ground Truth for Refined Captions:**
>
> Our refined captions are generated through a two-step process using GPT-4o (Line 234). First, we use regular expression matching to extract relevant contextual information from the original paper, specifically from sections closely associated with each figure (e.g., results, discussion sections). Second, GPT-4o performs rewriting and noise reduction by integrating this extracted context with the original figure and its raw caption. This approach corresponds to your category (3) (generated using LLMs), but is strictly grounded in category (1) (extracted paper text), with no independent human rewriting involved.
>
> **Ground Truth for QA Pairs and Reasoning Tasks:**
>
> These ground truth annotations are derived from the refined captions through a validated pipeline (Line 245 and Appendix B.6). LLMs first generate initial QA pairs and reasoning answers based on the structured information in refined captions. To ensure correctness, all generated content undergoes dual validation: first by multiple MLLMs (Section 3.3.1), then by human experts (Section 3.3.2). Only content passing both validation rounds is retained as final ground truth. This combines your category (3) (LLM-generated) with human expert oversight to ensure high reliability and eliminate errors.
>
> In summary, all ground truth is LLM-generated but rigorously validated through either source text grounding (for captions) or dual human-machine verification (for QA and reasoning tasks).
>
>
> > W2: The benchmark does not include human or expert performance.
>
> Thank you for this critical observation—we fully agree that human/expert baselines are essential to validate benchmark difficulty and substantiate our "graduate-level reasoning" claim. We have detailed the human performance baseline in Appendix J of the submission, which directly addresses your concerns.
>
> To rigorously assess benchmark difficulty and justify its educational relevance, we incorporated human expert evaluations in MSEarth-Bench-mini. We recruited three human experts—all Ph.D. students with specialized backgrounds in Earth sciences—to complete the tasks independently. Their average performance serves as the official human baseline, providing a concrete reference for comparing MLLM results.
>
> The expert team achieves an average accuracy of 87.00%, with performance ranging from 85.00% in Atmospheric Science to 92.86% in Ecology. They maintain exceptional performance across all subfields of Earth science. In contrast, the top-performing MLLM, o4-mini, only reaches 53.00% on average. Even its best domain performance of 71.43% in Ecology still falls 21.43 percentage points below the expert baseline. Other leading models show even larger gaps: gemini-2.5-pro achieves 51.33% and InternVL3-78B achieves 47.33%. This substantial gap between expert and MLLM performance confirms that MSEarth-Bench presents genuine challenges requiring domain expertise and advanced reasoning capabilities, validating its position as a graduate-level benchmark.
>
> **In the revised submission, we have incorporated this human expert baseline analysis into Section 4.3.3 of the main text for improved accessibility and prominence.**

---

> ### Author Response · Authors · 2025-11-19
> **Response to Reviewer nNLz (2/2)**
>
> > W3: The paper does not provide a deeper analysis of why models fail or which types of reasoning are most challenging.
>
>
> Thank you for this constructive feedback. We address model failure analysis through our domain-specific task design and targeted analyses, which provide actionable insights for model improvement.
>
> **Domain-Specific Classification for Actionable Insights:**
>
> We adopt domain-based categorization rather than generic reasoning skill taxonomies because Earth science reasoning demands are inherently tied to disciplinary knowledge. Following the approach of ScienceQA and MMMU, this classification enables more practical and granular differentiation for guiding domain-adapted model development.
>
> For example, Meteorology focuses on short-term weather-related reasoning (e.g., "synoptic-scale currents interacting with topography" in MCQ N0033), while Climatology emphasizes long-term trend analysis (e.g., "temperature increase across decades" in the MCQ N007963). Even for the same core skill (trend prediction), the disciplinary context (time scale, domain variables) creates distinct challenges that generic reasoning taxonomies miss.
>
> **We provide comprehensive performance breakdowns in Tables 17–20 of the revised submission, showing detailed accuracy across all Earth science subfields for every model.** These tables reveal consistent underperformance in specific subfields: Environmental Oceanography and Climatology show notably lower accuracy across most models. This granular data pinpoints clear, actionable weak points for future optimization—insights that generic reasoning classifications cannot provide.
>
> We identify specific failure modes and improvement strategies through three key findings:
>
> **Data-driven improvement:** Models trained on scientific papers, such as Intern-S1 and Intern-S1-mini, significantly outperform general-purpose models as shown in Table 2. This demonstrates that integrating domain-specific academic data is critical for bridging the reasoning gap.
>
> **Explicit reasoning enhancement:** Appendix K demonstrates that incorporating explicit reasoning prompts—such as step-by-step scientific deduction—significantly boosts model performance. This directly identifies the lack of structured scientific reasoning as a key failure point.
>
> **Targeted training validation:** Our RL-trained model Intern-S1-mini-MSEarth achieves substantial improvements, demonstrating that fine-tuning with MSEarth's domain-specific data combined with RL methods effectively enhances scientific reasoning capabilities. This provides an actionable training pipeline for future model development.
>
> These findings collectively provide both diagnostic insights into current model limitations and concrete pathways for improvement.

---

### Note · Authors · 2026-01-06

I have read and agree with the venue's withdrawal policy on behalf of myself and my co-authors.